# Changes in crop yields and their variability at different levels of global warming

**Sebastian Ostberg[1,2], Jacob Schewe[1], Katelin Childers[1], and Katja Frieler[1]**

[1]Potsdam Institute for Climate Impact Research, Telegrafenberg A31, 14473 Potsdam, Germany
[2]Geography Department, Humboldt-Universität zu Berlin, Berlin, Germany

**Correspondence:** Sebastian Ostberg (ostberg@pik-potsdam.de)

**Abstract.** An assessment of climate change impacts at different levels of global warming is crucial to inform the policy discussion about mitigation targets, as well as for the economic evaluation of climate change impacts. Integrated Assessment Models often use global mean temperature change ($\Delta$GMT) as a sole measure of climate change and, therefore, need to describe impacts as a function of $\Delta$GMT. There is already a well-established framework for the scalability of regional temperature and precipitation changes with $\Delta$GMT. It is less clear to what extent more complex, biological or physiological impacts such as crop yield changes can also be described in terms of $\Delta$GMT; even though such impacts may often be more directly relevant for human livelihoods than changes in the physical climate. Here we show that crop yield projections can indeed be described in terms of $\Delta$GMT to a large extent, allowing for a fast estimation of crop yield changes for emission scenarios not originally covered by climate and crop model projections. We use an ensemble of global gridded crop model simulations for the four major staple crops to show that the scenario dependence is a minor component of the overall variance of projected yield changes at different levels of $\Delta$GMT. In contrast, the variance is dominated by the spread across crop models. Varying $CO_2$ concentrations are shown to explain only a minor component of crop yield variability at different levels of global warming. In addition, we find that the variability of crop yields is expected to increase with increasing warming in many world regions. We provide, for each crop model, geographical patterns of mean yield changes that allow for a simplified description of yield changes under arbitrary pathways of global mean temperature and $CO_2$ changes, without the need for additional climate and crop model simulations.

## 1 Introduction

Climate change exerts a substantial and direct impact on food security and hunger risk by altering the global patterns of precipitation and temperature which determine the location of arable land (Parry et al., 2005; Rosenzweig et al., 2014) as well as the quality (Müller et al., 2014) and quantity (Müller and Robertson, 2014; Lobell et al., 2012; van der Velde et al., 2012) of crops comprising most of the world food supply. By itself, climate change is expected to reduce global production of the four major crops wheat, maize, soy and rice on current agricultural areas (e.g., Rosenzweig et al., 2014; Challinor and Wheeler, 2008; Peng et al., 2004). Facing an increasing food demand due to population growth and economic development, these reductions will have to be compensated by 1) the direct physiological impacts of increased atmospheric $CO_2$ concentrations (Kimball, 1983), which are beyond local human control; as well as 2) advances in agricultural management (e.g. fertilizer input or irrigation), technology, and breeding (Jaggard et al., 2010) or 3) expansion of agricultural land (Frieler et al., 2015; Smith et al., 2010).

In conjunction with these long term changes, global warming is also expected to contribute to an increase in the frequency and duration of extreme temperatures and precipitation (droughts, floods, and heat waves), which may increase the near term variability of crop yields and trigger short term crop price fluctuations (Brown and Kshirsagar, 2015; Mendelsohn et al., 2007; Tadesse et al., 2014).

Anthropogenic emissions of greenhouse gases are expected to influence crop yields via several pathways. On the one hand, the associated climatic changes will modify the length of the growing season (Eyshi Rezaei et al., 2014), water availability, and heat stress (Lobell et al., 2012; Müller and Robertson, 2014; Schlenker and Roberts, 2009); and on

the other hand, higher concentrations of atmospheric $CO_2$ are expected to increase the water use efficiency in C3 (e.g. wheat, rice, soy) and C4 (maize) crops, and enhance the rate of photosynthesis in C3 crops (Darwin and Kennedy, 2000). Global Gridded Crop Models (GGCMs) are particularly designed to account for these effects. They provide a complex process-based implementation of our current understanding of the mechanisms underlying crop growth, and are the primary tool for crop yield projections (e.g., Rosenzweig et al., 2014) which in turn are a prerequisite for assessing potential changes in prices (Nelson et al., 2014) and food security (Parry et al., 2005). However, these process-based crop yield projections rely on spatially explicit realizations of the driving weather variables such as temperature, precipitation, radiation, and humidity, often at daily resolution, as provided by computationally expensive Global Climate Model (GCM) simulations. The GGCMs themselves also require significant computational capacity. These requirements generally limit the number and length of emission scenarios that can be simulated.

The so-called pattern scaling approach is a well-established method to overcome these limits. Output from GCMs has been shown to be, to some extent, scalable to different global mean temperature (GMT) trajectories not originally covered by GCM simulations (Santer et al., 1990; Mitchell, 2003; IPCC-TGICA, 2007; Giorgi, 2008; Solomon et al., 2009; Frieler et al., 2012; Heinke et al., 2013). Scaled climate projections have also been used as input for different impact models (Ostberg et al., 2013; Stehfest et al., 2014) to achieve greater flexibility in terms of the range of emission scenarios considered in climate impact studies.

Building upon such a framework, we present a method to extend the capacity of crop yield impact projections by relating simulated crop yield changes to two highly aggregated quantities – global mean temperature change ($\Delta$GMT) and atmospheric $CO_2$ concentration ($pCO_2$) – by means of simplified function. $\Delta$GMT and $pCO_2$ are standard outputs of reduced-complexity climate models, which – while lacking the spatial resolution of complex GCMs – allow for highly efficient climate projections for any emissions scenario by emulating the response of the complex models (Meinshausen et al., 2011). Here "emulating" means that the simplified representation is designed to reproduce the global response of the complex model for the originally simulated scenarios but also allows for its inter- or extrapolation to other scenarios. We test to what extent crop yield changes, as one example of climate change impacts, can be described directly in terms of GMT and $pCO_2$ changes. Our approach is different from other emulators which use spatially explicit climate projections as input for the simplified functions (Oyebamiji et al., 2015; Blanc, 2017). While these approaches only emulate the responses of the complex crop model, the approach presented here implicitly provides a simplified description of both the GCMs' regional patterns of climate change and the associated response of the crop models. Such an approach provides

high computational efficiency, making it applicable, for example, in Integrated Assessment Models. In principle, other emulators could be used in this setting, however requiring an additional step of first scaling the climatic changes to the specific emission scenario.

The emulator introduced here allows for multi-crop-model projections for arbitrary emission scenarios as long as crop-model ensemble projections are available for a limited set of scenarios. This offers a practical way of keeping track of a relevant but often-ignored source of uncertainty which is manifested in the considerable spread across different crop models and other process-based impact models (Rosenzweig et al., 2014; Schewe et al., 2014). This uncertainty is particularly critical when estimating socio-economic consequences (e.g., Nelson et al., 2014).

We test the approach using an ensemble of yield projections of the four major crops maize, rice, soy, and wheat, generated within the first phase ("Fast Track") of the Intersectoral Impact Model Intercomparison Project (ISIMIP, Warszawski et al., 2014). For a number of $\Delta$GMT intervals we compare the spread in yield outcomes induced by the choice of emission scenario with that induced by the choice of GGCM and GCM, respectively. A low scenario-induced spread means that GCM- and GGCM-specific yield projections can be approximated by a simplified relationship with global mean temperature change without accounting for the underlying emission scenario, which is a prerequisite to applying the simplified relationship to other emission scenarios. The test is done at each grid point and separately for simulations of purely rain-fed yields and fully irrigated yields. Multi-model ensembles in the ISIMIP data archive provide a uniquely broad suite of crop yield simulations over a wide range of crops, $CO_2$ concentrations, and irrigation options encompassing output from five GGCMs, forced with output from up to five GCMs, and four Representative Concentration Pathways (RCPs, van Vuuren et al., 2011).

In Section 2 we describe the ISIMIP data and the methods used to test for scenario dependence and adjustment for different levels of $pCO_2$. Section 3 is dedicated to the presentation of the projected average changes in crop yields at different levels of global warming and an attribution of the variance of these long term changes to different sources of uncertainty, i.e., different GCMs, different GGCMs, and different emission scenarios (subsection 3.1). In addition, we test to what degree the scenario-dependence of crop yields at a specific level of global warming can be explained by different levels of $pCO_2$ (subsection 3.2). Finally, we provide individual maps of yield changes at different levels of GMT and the additional effect of variations in $pCO_2$ at the respective GMT levels. We propose three methods to generate these patterns based on the available complex model simulations, and describe the related approaches to estimate GGCM- and GCM-specific yield changes for new $\Delta$GMT trajectories not originally covered by GCM-crop-model simulations. In Section 4 we present a quantification of the projection errors as

compared to actual simulations by the complex gridded crop models. Finally, in Section 5 we quantify the residual inter-annual variance of the simulated crop yields in terms of GMT change across all combinations of crop and climate models.
[5] Section 6 provides a summary.

## 2 Data and Methods

### 2.1 Crop yield simulations

We use projections from five different GGCMs (GEPIC, LPJ-GUESS, LPJmL, PEGASUS, and pDSSAT) that par-[10] ticipated in the first phase of ISIMIP (Rosenzweig et al., 2014; Warszawski et al., 2014) in order to test for a dependence of projected yield changes on the GMT pathway (see Table 1 for their basic characteristics). Each crop model was forced by climate projections from five differ-[15] ent GCMs (HadGEM2-ES, IPSL-CM5A-LR, MIROC-ESM-CHEM, GFDL-ESM2M, NorESM1-M) generated for four RCPs (RCP2.6, RCP4.5, RCP6.0, RCP8.5) in the context of the Coupled Model Intercomparison Project, phase 5 (CMIP5, Taylor et al., 2012). CMIP5 was an effort by the [20] climate modelling community to provide a new suite of climate simulations in time for the Intergovernmental Panel on Climate Change (IPCC) Fifth Assessment Report (AR5). The RCPs cover the range from climate mitigation (RCP2.6, RCP4.5) to business-as-usual (RCP6.0) and high emissions [25] scenarios (RCP8.5). Climate projections have been bias-corrected to better match observed historical averages of the considered climate variables. For the future, the bias-correction preserves absolute changes in monthly temperature and relative changes in monthly values of the other vari-[30] ables simulated by the GCMs while also correcting the daily variability about the monthly mean (Hempel et al., 2013). Separate simulations are available for each of the four major crops: wheat, maize, rice and soy, on a global 0.5 x 0.5 degree grid, covering the time period from 1971–2099. The consid-[35] ered crop is assumed to grow everywhere on the global land area, only restricted by soil characteristics and climate but independent of present or future land use patterns ("pure crop" simulations). Each model has provided a pair of simulations ("runs") for each climate change scenario: 1) a rain-fed run [40] and 2) a full-irrigation run assuming no water constraints. This design provides full flexibility with regard to the application of future land use and irrigation patterns. While the crop yield ($Y_{varCO2}$) in "default" simulations accounts for the fertilization effects due to the increasing levels of [45] pCO$_2$, the ISIMIP setting also includes a sensitivity experiment where the crop models were forced by the same climate change projections but pCO$_2$ was kept fixed at a "present day" reference level that differs from GGCM to GGCM (see Table 1). We will refer to this run as "fixed CO$_2$" run and [50] indicate the associated crop yields by $Y_{fixedCO2}$. As a special case, the "default" simulations for pDSSAT do not use annual pCO$_2$ changes. Instead, pCO$_2$ was changed every 30

years using the average pCO$_2$ of the respective 30 year time slice.

### 2.2 Effect of temperature change

We analyse the dependence of yield changes on $\Delta$GMT separately for rain-fed and full-irrigation simulations, and for each crop. While yields in a given grid cell of course depend on the local temperature, long-term changes in local temperature are in turn a manifestation of global greenhouse-[60] gas related warming (Frieler et al., 2012). The aim here is testing to what extent local long-term changes in yields can be described in terms of a single global measure of warming, $\Delta$GMT. Since the time of attaining a given $\Delta$GMT differs between GCMs and scenarios, we group all available [65] data into $\Delta$GMT intervals (bins) separated by 0.5°C steps with 0.5°C width ($\pm$0.25°C around the central temperature), where $\Delta$GMT is calculated relative to the present day (1980–2010 average) reference level. For all annual data falling into a given interval and at each grid point we apply a separate [70] one-way analysis of variance (ANOVA fixed effects model) to individually calculate the yield variance explained by 1) different GGCMs, 2) the GCMs, and 3) the RCPs. The quantification of the RCP-dependence of the relationship between global warming and yield change is limited to warming lev-[75] els up to 2 to 3°C above present depending on the GCM because only one RCP (RCP8.5) reaches temperatures above this threshold. However, we also provide the patterns of yield change for the higher concentration scenario. In the main text, all figures except 9 & 10 refer to a $\Delta$GMT level of [80] 2.5°C, and all figures except 3, 4, and 11 refer to crop model simulations driven by HadGEM2-ES climate. See Figure 1 for the years associated with $\Delta$GMT=2.5°C in HadGEM2-ES. The Supplement contains analogous figures for other GMT levels and GCMs. [85]

We do not impose a specific functional relationship between GMT change and change in crop yields. Yield change for any GMT level between the central levels of the considered bins could be derived by a simple linear interpolation between the patterns of neighbouring bins but without as-[90] suming a linear relationship between global mean warming and yield change across the full range of warming.

### 2.3 Effect of pCO$_2$ change

The direct effect of CO$_2$ fertilization on crop yields is expected to introduce some scenario dependence in the rela-[95] tionship between GMT change and yield change. We test to what degree the scenario dependence of the relationship can be explained by introducing pCO$_2$ as an additional predictor for within-bin fluctuation of yields. To this end, we evaluate two different approaches to estimate the direct CO$_2$ effect on [100] crop yields within the different GMT bins, described in detail below. The two approaches differ in terms of the crop model simulations that they require: approach (a) only requires the

**Table 1.** Basic crop model characteristics with respect to 1) the implementation of $CO_2$ fertilization effect (as affecting radiation use efficiency (RUE), transpiration efficiency (TE), leaf level photosynthesis (LLP), or canopy conductance (CC)), 2) the accounting for nutrient constraints and associated assumption with respect to fertilizer application (N = nitrogen, P = phosphorus, K = potassium), 3) implemented adaptation measures.

| Model | $CO_2$ fertilization | Nutrient limitation | Adaptation |
|---|---|---|---|
| GEPIC (Liu et al., 2007; Liu, 2009) | RUE, TE pCO2 of the fixed $CO_2$ run: 364 ppm | flexible N application up to an upper national application limit according to FAO FertiStat database (FAO, 2007), fixed present-day P application rates following FertiStat. | decadal adjustment of planting dates (incl. switch between winter and spring wheat); total heat units to reach maturity remain constant |
| LPJ-GUESS (Lindeskog et al., 2013) | LLP, CC pCO2 of the fixed $CO_2$ run: 379 ppm | no consideration of soil nutrient limitation | adjustment of total heat units to reach maturity based on the average climate during the preceding 10 years to keep growing season length constant |
| LPJmL (Bondeau et al., 2007) | LLP, CC pCO2 of the fixed $CO_2$ run: 370 ppm | no consideration of soil nutrient limitation | fixed sowing dates (Waha et al., 2012); total heat units to reach maturity remain constant |
| PEGASUS (Deryng et al., 2011) | RUE, TE pCO2 of the fixed $CO_2$ run: 369 ppm | fixed N, P, K application rates (IFA, 2002) | adjustment of planting dates; variable heat units to reach maturity |
| pDSSAT (Jones et al., 2003; Elliott et al., 2014) | RUE, LLP, CC pCO2 of the fixed $CO_2$ run: 330 ppm | fixed N present-day application rates | no adjustment of planting dates; total heat units to reach maturity remain constant |

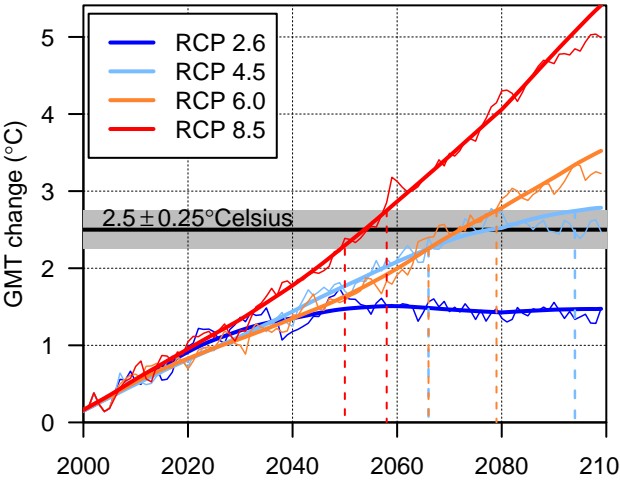

**Figure 1.** GMT projections from HadGEM2-ES for the four RCPs. The horizontal line and shading indicate the 2.5°C bin. The original annual GMT values (thin lines) are smoothed (thick lines) in order to obtain a contiguous time interval for each $\Delta$GMT bin. The smoothing is based on a Singular Spectrum Analysis with a time window of 20 years (R-Package Rssa, Korobeynikov, 2010; Golyandina and Korobeynikov, 2014; Golyandina et al., 2015). Years where the thick line falls within the shaded area are associated with $\Delta$GMT = 2.5°C, and the corresponding time interval is delineated by the dashed vertical lines.

default crop yield simulations with increasing pCO$_2$ whereas approach (b) requires a pair of simulations with increasing pCO$_2$ and with fixed pCO$_2$ at present-day reference level.

### 2.3.1 Approach (a)

For all years falling into a specific $\Delta$GMT bin, approach (a) fits the following linear regression model to the response of yields in the default simulation to the increase in pCO$_2$:

$$\Delta Y_{\text{varCO2}}(i,t) = \Delta Y_{\text{clim}}(i) + a_1(i) \cdot (\text{pCO2}(t) - 370\text{ppm}) + \epsilon(i,t), \qquad (1)$$

where $\Delta Y_{\text{varCO2}}(i,t)$ is the absolute yield change in grid point $i$ and year $t$ with respect to the historical reference period (1980–2010) and pCO2$(t)$ is the atmospheric $CO_2$ concentration of the corresponding year. In this statistical model, two parameters are determined by regression: $\Delta Y_{\text{clim}}(i)$ represents an estimate of the purely climate-induced yield change at the respective bin temperature, but assuming a fixed year-2000 pCO$_2$ of 370 ppm (i.e. without $CO_2$ fertilization), and $a_1(i)$ represents the added effect of $CO_2$ fertilization. Finally, $\epsilon(i,t) \backsim N(0, \sigma^2)$ represents the residual error.

### 2.3.2 Approach (b)

Approach (b) fits the following linear regression model to the yield difference between the default and fixed-$CO_2$ simula-

tion for all years falling into a specific $\Delta$GMT bin:

$$Y_{\text{varCO2}}(i,t) - Y_{\text{fixedCO2}}(i,t) =$$
$$a_1(i) \cdot (\text{pCO2}(t) - \text{pCO2}_{\text{ref}}) + \epsilon(i,t), \quad (2)$$

where $Y_{\text{varCO2}}(i,t)$ and $Y_{\text{fixedCO2}}(i,t)$ is the absolute yield in grid point $i$ and year $t$ of the default and fixed-$CO_2$ simulation, respectively, $\text{pCO2}(t)$ is the atmospheric $CO_2$ concentration of the default simulation during the respective year and $\text{pCO2}_{\text{ref}}$ is the crop-model specific $\text{pCO}_2$ value of the fixed-$CO_2$ simulation (see Table 1). In this statistical model, $a_1(i)$ is determined by regression and represents the $CO_2$ fertilization effect, and $\epsilon(i,t) \frown N(0,\sigma^2)$ represents the residual error. No intercept is estimated in this model because yields from the default and fixed-$CO_2$ runs are expected to be identical if $\text{pCO2(t)} = \text{pCO2}_{\text{ref}}$. The purely climate-induced yield change at a fixed year-2000 $\text{pCO}_2$ of 370 ppm $\Delta Y_{\text{clim}}(i)$ can then be derived as:

$$\Delta Y_{\text{clim}}(i) = \Delta Y_{\text{fixedCO2}}(i) + a_1(i) \cdot (\text{pCO2}_{\text{ref}} - 370\text{ppm}), \quad (3)$$

where $\Delta Y_{\text{fixedCO2}}(i)$ is the average yield change in the respective warming bin of the fixed $CO_2$ simulation with respect to the historical reference period and $a_1(i) \cdot (\text{pCO2}_{\text{ref}} - 370\text{ppm})$ corrects for the different $\text{pCO2}_{\text{ref}}$ used by each GGCM.

## 2.4   Emulator of temperature and $CO_2$ effects

Based on the spatial patterns of purely climate-induced yield change $\Delta Y_{\text{clim}}(i)$ and added $CO_2$ fertilization effect $a_1(i)$, which are derived separately for each rain-fed and irrigated crop and specific to each crop model and GCM, we propose the following two-step interpolation method to compute crop yield changes for any given pair of $\Delta$GMT and $\text{pCO}_2$, using either the coefficients from approach (a) or (b):

1.  linear interpolation of $\Delta Y_{\text{clim}}(i)$ between the two neighbouring $\Delta$GMT bins to the desired $\Delta$GMT value,

2.  addition of the $CO_2$ pattern described by $a_1(i) \cdot (\text{pCO2} - 370\text{ppm})$, where $a_1(i)$ is also interpolated linearly between the respective coefficients from the neighbouring $\Delta$GMT bins.

The application of these two steps using coefficients from method (a) above will be called emulator approach (a); their application using coefficients from regression method (b) will be called emulator approach (b). In addition, we propose a third, very basic emulator approach (c) where the yield change for any given $\Delta$GMT is derived from a simple linear interpolation of the average yield change in the neighbouring warming bins of the default simulations $\Delta Y_{\text{varCO2}}(i)$ with respect to the historical reference period, without using the associated $\text{pCO}_2$ as additional predictor.

The linear interpolation of any of the previous coefficients between two neighbouring warming bins is illustrated for a $\Delta$GMT of 2.3°C as follows:

$$\text{coef}(i, 2.3°\text{C}) = (1 - \delta) \cdot \text{coef}(i, 2°\text{C}) + \delta \cdot \text{coef}(i, 2.5°\text{C}),$$
$$\delta = (2.3°\text{C} - 2°\text{C})/(2.5°\text{C} - 2°\text{C}), \quad (4)$$

where coef can be $\Delta Y_{\text{clim}}(i)$, $a_1(i)$, or $\Delta Y_{\text{varCO2}}(i)$.

Using GGCM projections for the HadGEM2-ES climate input to train the emulators, we test which of the emulator approaches, (a), (b) or (c), provides the best reproducibility for yield changes simulated under the four RCPs (Section 4). While approach (b) requires a pair of crop model simulations – one with time-varying $\text{pCO}_2$ and one with fixed present-day $\text{pCO}_2$ – approach (a) and (c) only require the default simulations with time-varying $\text{pCO}_2$. Thus, a comparison of the three approaches could provide some important guidance regarding future crop model experiments required to allow for the proposed highly efficient emulation of crop model simulations. Since simulated crop yields are subject to considerable inter-annual variability, we also test what effect the amount of available training data has on the reliability of the derived regression coefficients. For that purpose, we train the emulators using either all available simulation data from the four RCPs or only simulation data from RCP8.5 and compare the fraction of the land surface for which derived fits are statistically significant as well as the difference between simulated and emulated yield changes. Due to the 30-year time slices of constant $\text{pCO}_2$ used by pDSSAT in the default run, approach (a) cannot be applied to this model using only RCP8.5 data. Since only RCP8.5 reaches $\Delta$GMT$> 3.5°$C this limits the temperature range of emulator approach (a) for pDSSAT even when using all available training data.

We evaluate and compare the performance of the three emulator approaches at the grid scale as well as the scale of large regions. Grid point yields (in t/ha) are multiplied by the fixed year-2000 crop-specific growing area from the MIRCA2000 dataset (Portmann et al., 2010) to derive regional total crop production (in t). MIRCA2000 provides gridded growing areas for a total of 26 rain-fed and irrigated crops based on a combination of census, remote sensing and other geographic data sources.

## 3   Mean Yield Change with Global Mean Temperature Change

### 3.1   Patterns of relative changes at different levels of global warming and main sources of variance

In general, increasing global mean temperatures correspond to an expansion of arable land to higher latitudes with concurrent yield reductions in equatorial regions. The highest positive changes in projected yields under rain-fed conditions at 2.5°C $\Delta$GMT are typically in the northern high latitudes and mountainous regions for all crops (Figure 2 for wheat,

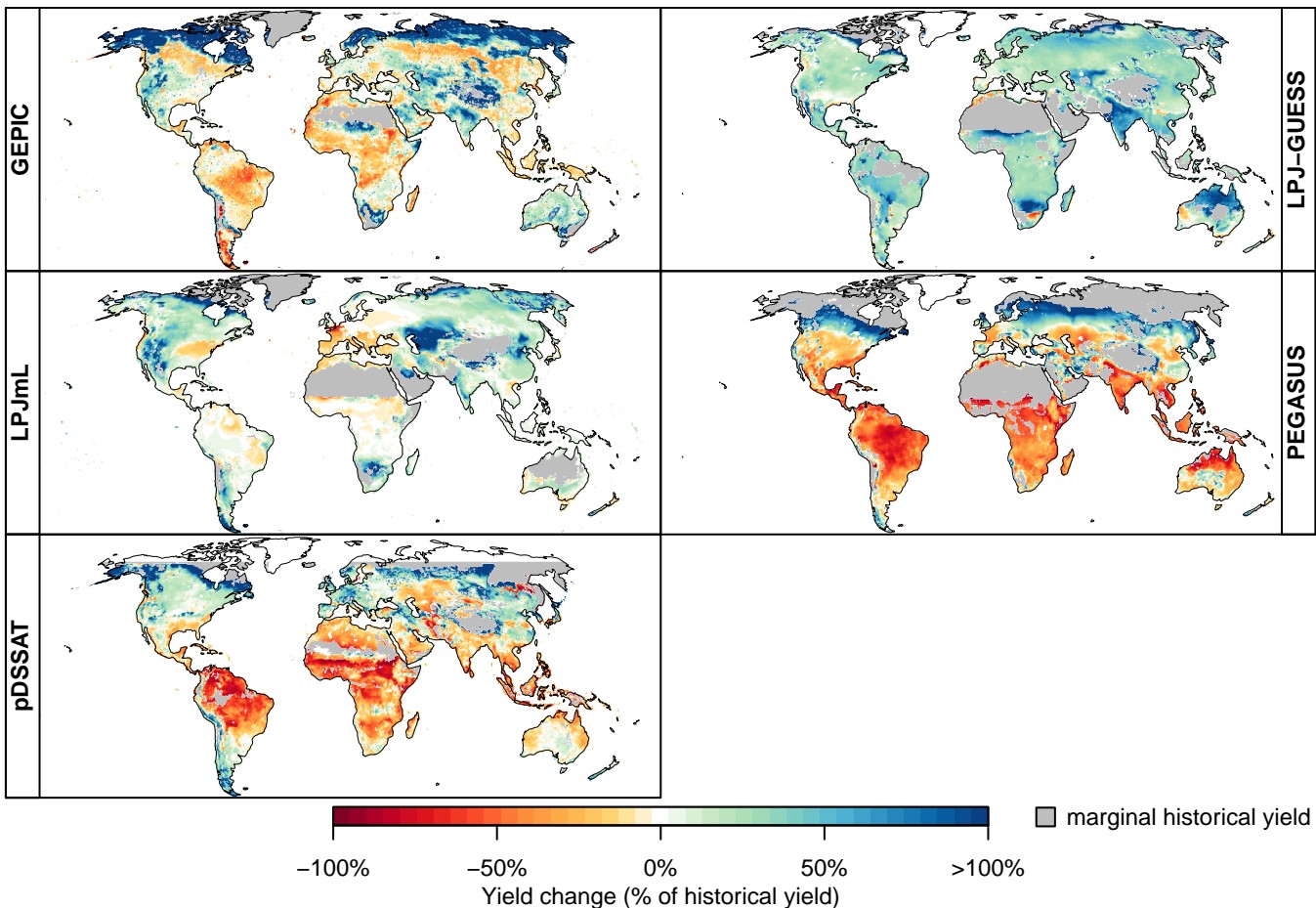

**Figure 2.** Average wheat yield change at ΔGMT=2.5°C as a percentage of the mean historical yield (1980–2010 average) under rain-fed conditions for each crop model forced by HadGEM2-ES. The average is calculated across all RCPs which reach the global mean warming interval from 2.25 to 2.75°C, namely RCP4.5, RCP6.0, and RCP8.5. Note that pDSSAT is run over a limited domain excluding areas north of 60°N. Regions with marginal historical yields (defined as lying below the 2.5% quantile of historical yields on year-2000 cropland) are masked to avoid exaggerated relative yield increases. Analogous figures for different crops, for irrigated conditions, as well as for absolute yield change (in t/ha) are available in the Supplement.

figures for other crops in the Supplement). These locations were previously inhibited by a short growing season, which extends with increasing air temperature (Ramankutty et al., 2002). Yield gains also occur over previously moisture lim-
5 ited regions, such as the northwestern U.S. and north-eastern China, in agreement with the findings of Ramankutty et al. (2002). In contrast, near the equator most crop yields decrease, especially maize and wheat. Since most cultivated land currently lies in low and middle latitudes, potential yield
10 changes in those regions contribute a higher relative importance for today's food production system than changes in high latitudes.

While variations exist in the magnitude of projected yield changes, there is a high degree of consistency in the direction
15 of yield change across ensemble members, especially over the high latitudes, where most of the largest projected yield changes occur, but where yields are in general smaller (Fig-

ure 3). Utilizing output from all available combinations of GCM, GGCM, and RCP scenario, more than three-quarters of the ensemble members indicate increasing crop yields 20 over the upper mid latitudes in the northern hemisphere for all crops at 2.5°C.

The simulated yield values at each grid point and within each GMT bin are subject to variation due to the selection of impact model, GCM forcing, and emissions scenario. When 25 considering all of these factors, the variance attributable to the impact model selection is much greater than that associated with the GCM or scenario choice in most regions (Figure 4). This holds for rain-fed as well as irrigated simulations. The predominance of the impact model component in 30 total variance is particularly evident in the middle to high latitudes for all four considered crops, where impact model variance accounts for up to 90% of the grid point variance at 2.5°C.

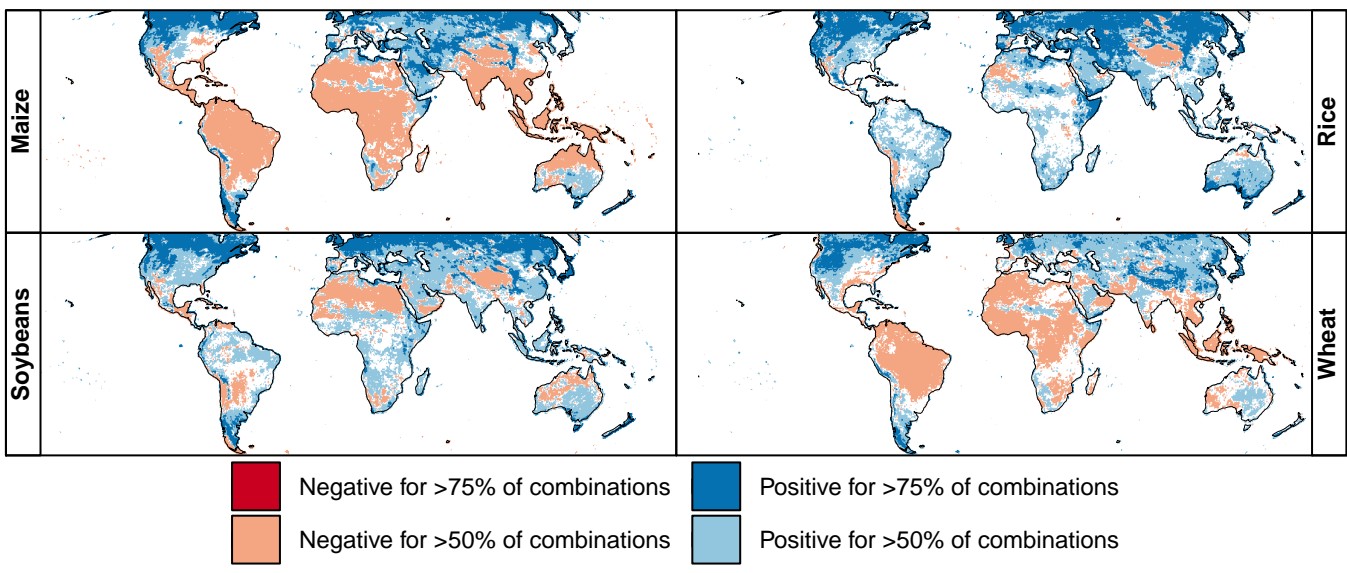

**Figure 3.** Percentage of crop model simulations (combination of a single GCM, GGCM, and RCP scenario) indicating an increase (blue) or decrease (red) in yield of greater than 5% at each grid point at $2.5 \pm 0.25°C$ $\Delta$GMT as compared to the historical period for maize, rice, soybeans, and wheat under rain-fed conditions. White indicates either a less than 5% change or disagreement between the models in the direction of yield change. Note that only four out of five GGCMs provided results for rice. An analogous figure for irrigated conditions is available in the Supplement.

## 3.2 Direct impacts of increasing $pCO_2$

In addition to air temperature warming, $pCO_2$ has a direct influence on crop yields. As it varies within the different $\Delta$GMT bins, it is expected to induce part of the fluctuations of the yield changes at given GMT levels. We find that this $CO_2$ effect shows little scenario dependence (see Figure 5 for the global average effect within the LPJmL simulations at $\Delta$GMT=2.5°C), consistent with a short response time of plants to $pCO_2$ changes. As expected, the $CO_2$-induced yield differences increase with heightened atmospheric $CO_2$ level under all emissions scenarios, implying a stronger $CO_2$ fertilization impact with increased $pCO_2$.

At the grid point level, two approaches have been used to separate purely climate-change-induced from $CO_2$-induced yield change (following Equation 1 to Equation 3). Figure 6 shows the climate-change-induced yield change at $\Delta$GMT=2.5°C for LPJmL under rain-fed conditions, using all available runs that fall into the warming bin to estimate $\Delta Y_{\mathrm{clim}}(i)$. Figures for irrigated conditions and the other GGCMs are available in the Supplement. The two methods result in broadly similar patterns, with yield increases in the upper mid- and high latitudes, mixed regions with decreases and increases in the lower mid-latitudes and mostly decreases in the tropics. However, the magnitude of change differs between the two approaches: approach (a) generally estimates larger changes outside the tropics while yield decreases in the tropics are larger in approach (b). There are also some regions where both approaches disagree regarding the direction of change, such as the high latitudes of both Western

North America and Eastern Russia for wheat and parts of Southeast and South Asia for all crops. Patterns of climate-induced yield change match better between both approaches under irrigated conditions (see Supplement).

In GEPIC, both approaches disagree on the direction of change for maize yields over large parts of Europe. In LPJ-GUESS, both approaches disagree on the direction of change in most of the tropics for all crops. While tropical yield change is predominantly negative in approach (b) mirroring results of the other crop models, approach (a) estimates mostly positive climate effects on tropical crops. In pDSSAT, approach (a) generally produces larger areas with negative yield change than approach (b). At the same time, positive yield effects in approach (a) have a larger magnitude than those in approach (b) in many regions. In PEGASUS, both approaches disagree on the direction of change over large parts of the U.S. for maize and soybeans, and large parts of China for wheat.

The estimates of $CO_2$-induced yield change also differ between the two approaches (Figure 7 for LPJmL results under rain-fed conditions). We expect $CO_2$ fertilization to have a positive or at least neutral effect on yields, and this is confirmed by approach (b) for all GGCMs and crops. Only GEPIC simulations show negative $CO_2$ effects on soybean and wheat yields in a few regions for approach (b). This can be explained by nutrient interactions in the model: $CO_2$ fertilization leads to yield increases first but also increases nutrient depletion in the soil compared to the fixed-$CO_2$ run. If fertilizer application is insufficient to replenish nutrient

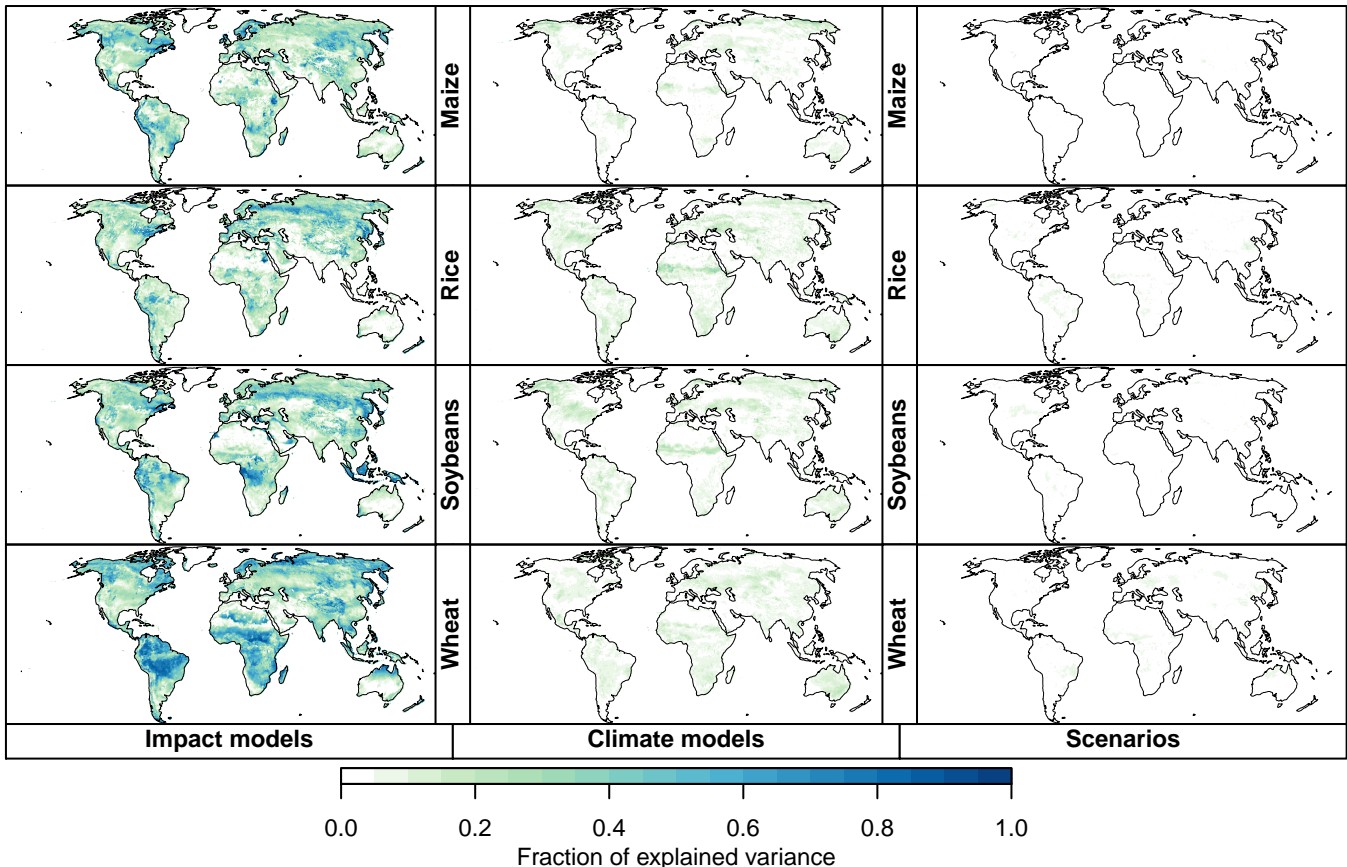

**Figure 4.** Fraction of total yield variance attributable to the impact models (GGCMs, left), climate models (GCMs, middle), and scenarios (RCPs, right) for each crop. Figure shown for rain-fed runs at $\Delta\mathrm{GMT} = 2.5 \pm 0.25^\circ\mathrm{C}$ warming; an analogous figure for irrigated runs is provided in the Supplement.

stocks this can lead to lower yields despite the beneficial effect of higher $pCO_2$. With approach (a), on the other hand, areas of negative estimated $CO_2$ effects are widespread in all GGCMs and all crops. Generally, the magnitudes of the estimated $CO_2$ effect are also much larger, often surpassing those of approach (b) even in regions where the direction of change matches. Given that approach (a) contradicts our expectation of how $CO_2$ fertilization should affect yields in many regions we conclude that approach (a) is not reliable in separating the effects of climate change on yield from those of $pCO_2$ change. By design, climate-induced and $CO_2$-induced yield changes add up to the full yield change (see Equation 1) which is why the difference between the patterns of estimated $CO_2$ effect explains why climate-change patterns from Figure 6 also differ substantially between both approaches in some regions. Approach (a) has a structural disadvantage to approach (b) in that it estimates both the climate-induced and $CO_2$-induced effect on yields from the same linear regression model (Equation 1). Besides changes in $pCO_2$ annual yields in each warming bin are subject to substantial inter-annual climate variability which means that individual years with a higher $pCO_2$ do not necessarily have

a higher yield. In contrast, approach (b) only estimates the $CO_2$-induced yield change from the regression model (Equation 2) while both the default and the fixed-$CO_2$ run are subject to identical climate variability. There is inter-annual variability in the $CO_2$-induced yield change as well (see Figure 5 for the global average effect), however, it is much smaller than the total yield variability. While approach (a) and (b) should provide similar estimates of the $CO_2$-induced yield change given a large sample, our sample size is limited by the number of years falling into each $\Delta\mathrm{GMT}$ bin (Table 2). This number varies between seven years in the 4.5 and 5.0°C bin and up to 66 years in the 1.0°C bin when yield data from all RCPs are used to train the emulator. The number of years varies between seven and 13 years if only data from RCP8.5 are used. Given the limited sample size and possibly large variability, the derived fits are often not statistically significant. For approach (a) we found that derived fits were rarely significant on more than 25% of the crop-specific growing area (Portmann et al., 2010) using a $p$-value of 0.05 (figure available in the Supplement). Values were even lower if only RCP8.5 was used for the regression. In contrast, fits derived by approach (b) were mostly statistically significant

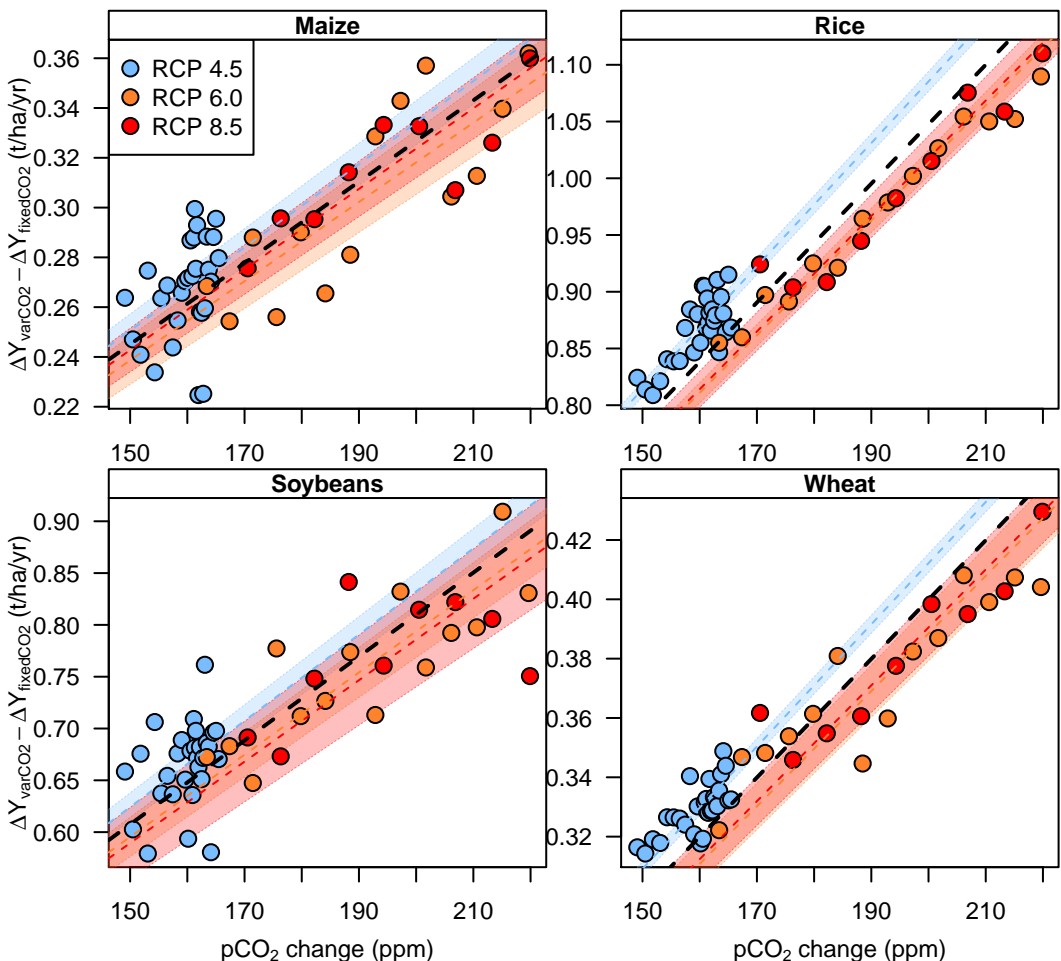

**Figure 5.** Difference in global mean yield change (sum of rain-fed and irrigated, and weighted by year-2000 growing areas) between the default ($Y_{varCO2}$) and fixed $CO_2$ simulations ($Y_{fixedCO2}$), for each crop over the range of $pCO_2$ associated with the $\Delta GMT = 2.5°C$ bin. Results are as simulated by LPJmL forced with output from HadGEM2-ES. Each color represents an emission scenario. Points mark individual years while dotted lines and shaded areas indicate the linear best fit and its 95% confidence interval for each scenario. The black dotted line indicates the linear best fit through all available scenarios. Analogous figures for other GGCMs and warming bins are available in the Supplement.

($p < 0.05$) on more than 70% of the growing area, often on more than 90% of the area. We also found only a small negative effect in terms of statistical significance if only RCP8.5 was used in approach (b).

## 4 Validation of three emulator approaches

Using GGCM projections for the HadGEM2-ES climate input, we test which of the approaches, (a), (b) or (c), provides the best reproducibility for all four RCPs. For that purpose, we apply each emulator with time series of $\Delta GMT$ and $pCO_2$ from the RCPs and compare emulated yield changes in each grid point and as well as total crop production for 10 large world regions to those simulated by the GGCM. For pDSSAT, the $pCO_2$ time series used in that model's default run is also used with the emulator.

Figure 8 shows results for the LPJmL model, when applying the emulators trained on all available data to reproduce rain-fed yields under RCP4.5. Figures for other RCPs, irrigated yields and other GGCMs are available in the Supplement.

Approach (a) generally leads to the largest differences relative to the simulated yield change (Figure 8, left column). In particular Maize, rice, and soybean yields are underestimated for much of North America, and overestimated in Europe, temperate South America, and Australia. Wheat yields are overestimated, e.g., in Canada.

Approach (b) also leads to some substantial deviations from the yields simulated by LPJmL, mainly in the northern hemisphere (Figure 8, middle column). Spatial patterns of over and underestimation are broadly similar to approach (a), but the magnitude of the difference is generally slightly

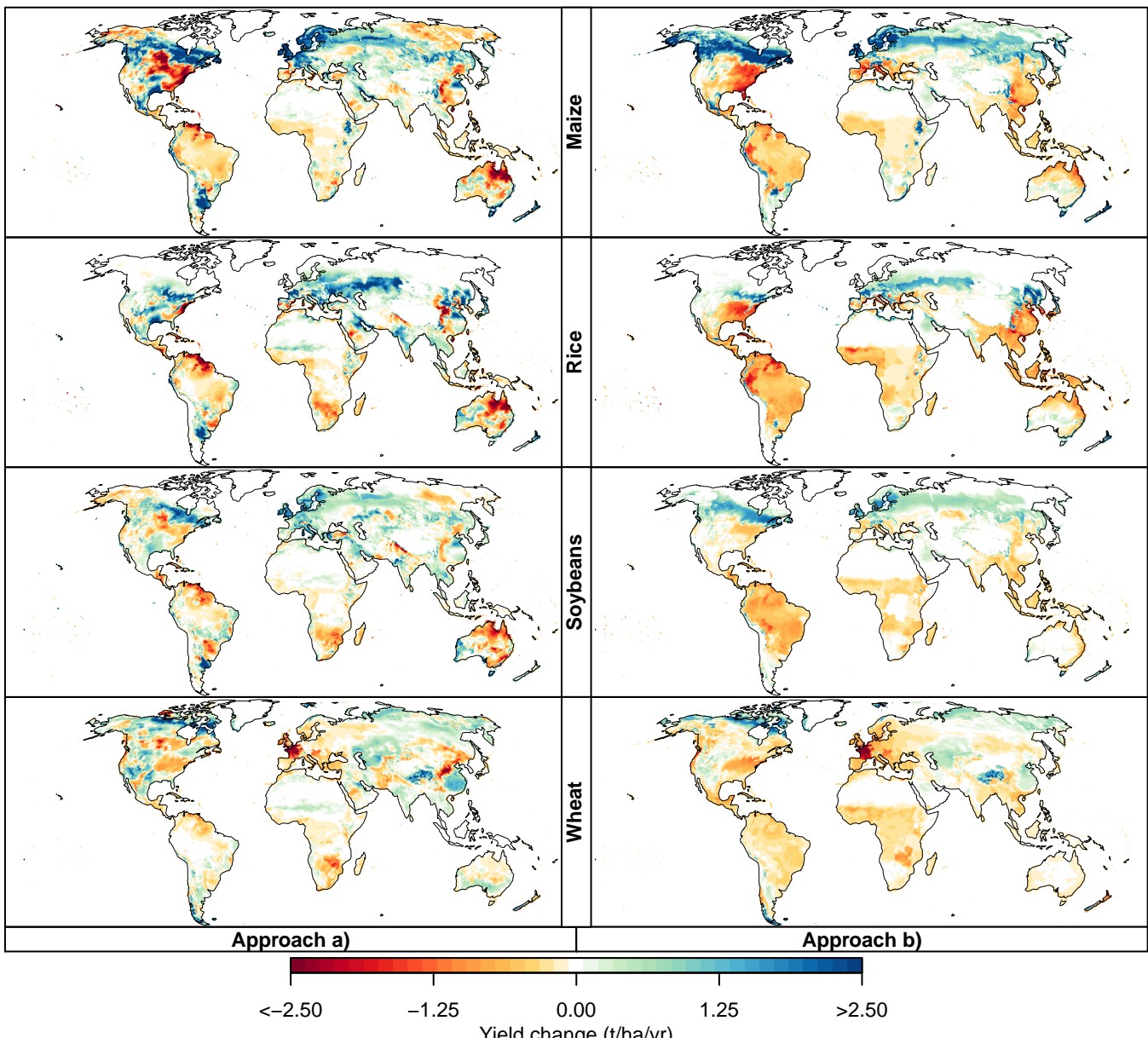

**Figure 6.** Climate change-induced yield changes at $\Delta$GMT=2.5°C of global warming and year 2000 $pCO_2$ level (370 ppm). Left column: Patterns of $\Delta Y_{\text{clim}}(i)$ derived at each grid point $i$ by approach (a) (see Equation 1). Right column: Corresponding patterns of $\Delta Y_{\text{clim}}(i)$, derived by approach (b) (see Equation 3). Both types of patterns are derived from LPJmL simulations forced by HadGEM2-ES assuming rain-fed conditions and expressed as absolute differences compared to the historical period (1980–2010). Rows: Different crop types. Analogous figures for irrigated conditions, for different GGCMs, and using relative instead of absolute yield changes are available in the Supplement.

**Table 2.** Number of years of yield data available in each $\Delta$GMT bin for HadGEM2-ES. Only RCP8.5 reaches warming levels above 3°C.

| Data used | $\Delta$GMT bin | | | | | | | | | |
|---|---|---|---|---|---|---|---|---|---|---|
| | 0.5°C | 1.0°C | 1.5°C | 2.0°C | 2.5°C | 3.0°C | 3.5°C | 4.0°C | 4.5°C | 5.0°C |
| all available scenarios | 47 | 66 | 44 | 38 | 52 | 20 | 8 | 8 | 7 | 7 |
| RCP8.5 only | 10 | 13 | 12 | 10 | 9 | 8 | 8 | 8 | 7 | 7 |

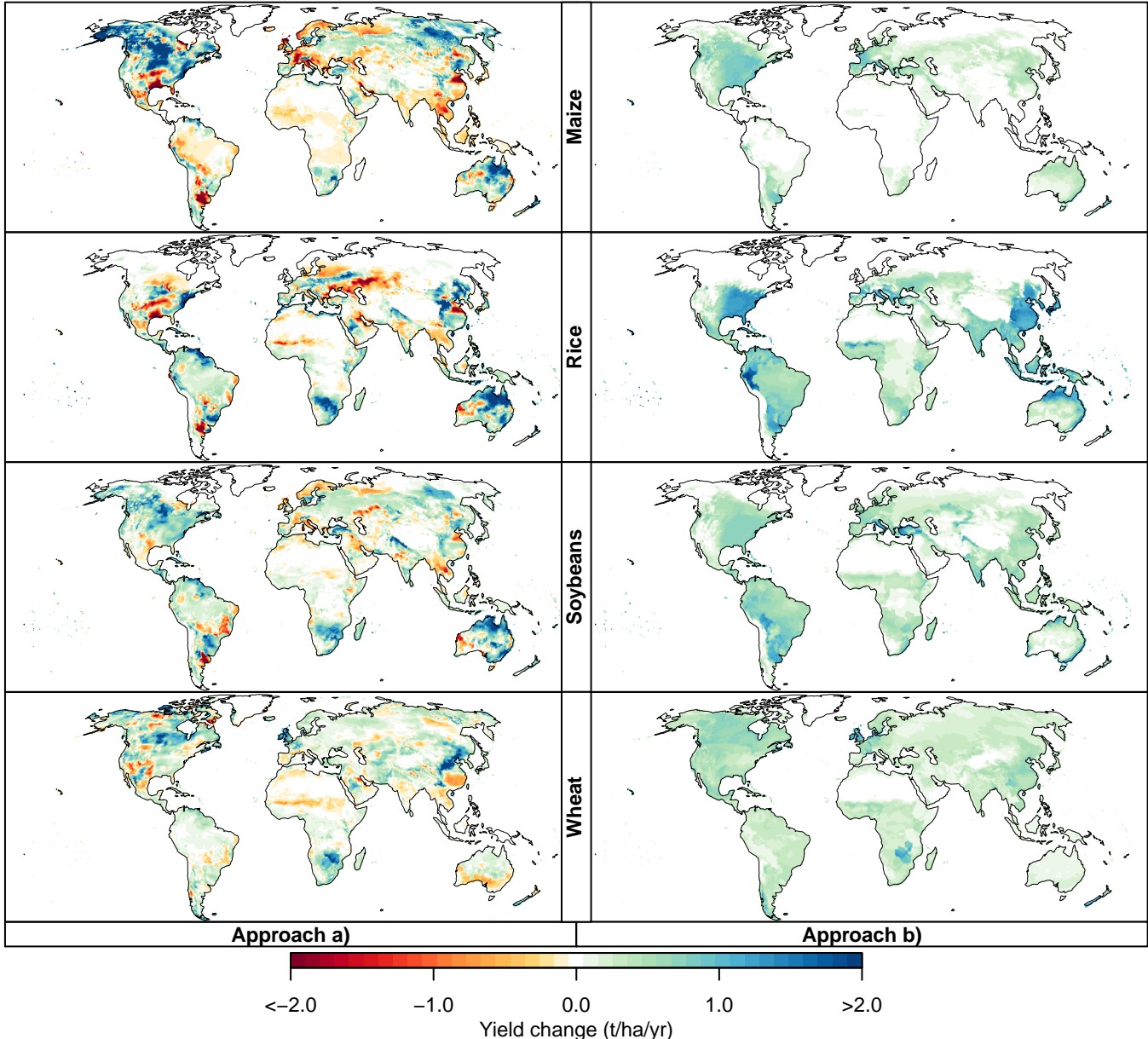

**Figure 7.** $CO_2$-induced yield changes at 2.5°C of global warming for LPJmL forced by HadGEM2-ES assuming rain-fed conditions. Analogous to Figure 6, but showing the scaling coefficients $a_1(i)$ from approach (a) (left column) and approach (b) (right column), multiplied by the average $pCO_2$ change compared to year 2000 (370 ppm) across all years falling into the GMT bin. Rows: Different crop types. Analogous figures for irrigated conditions, for different GGCMs, and using relative instead of absolute yield changes are available in the Supplement.

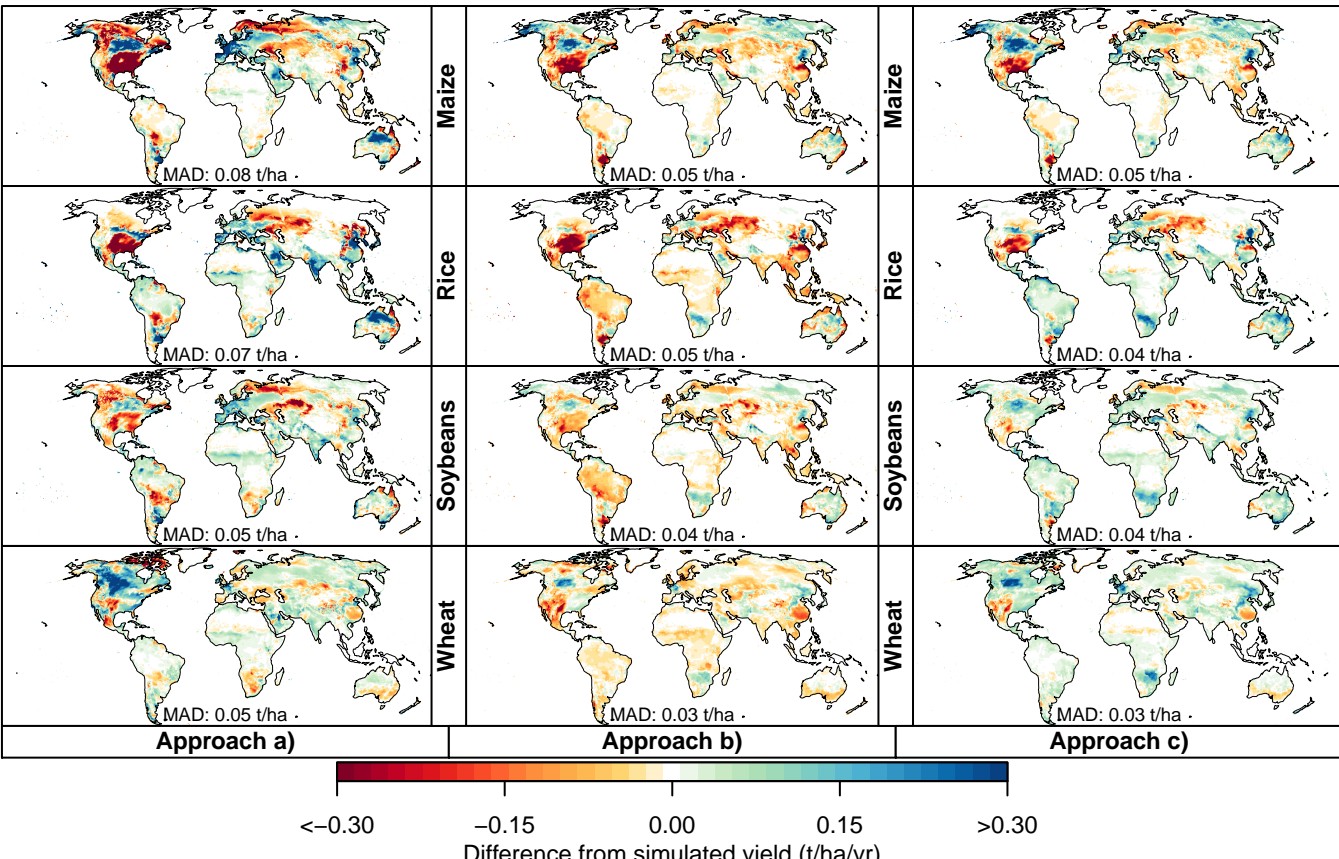

**Figure 8.** Validation of the three emulator approaches. Maps show the difference (emulated minus simulated) between the simulated LPJmL yields forced by HadGEM2-ES climate for RCP4.5 under rain-fed conditions, averaged over all years falling into the ΔGMT bin of 2.5°C (2066–2094), and the emulated yields for the same years based on approach (a) (left column), approach (b) (middle column), and approach (c) (right column). Rows: Different crops. MAD: mean absolute difference, regardless of sign, averaged across all grid points. Analogous figures for irrigated conditions and for different GGCMs are available in the Supplement.

lower. In the tropics, approach (b) often leads to a higher deviation from the simulated yields than approach (a), particularly for rice and soybeans in South America.

Finally, approach (c) leads to a similar pattern of deviations from the simulated yields as approach (b) for maize (Figure 8, right column). For the other crops, approach (c) often leads to an overestimation of yields whereas approach (b) tends to underestimate simulated yields. The mean absolute deviation between emulated and simulated yields (designated as MAD in Figure 8) is similar for approach (b) and (c). Approach (c) performs slightly better than approach (b) for rice, and both approach (b) and (c) perform better than approach (a) for all four crops. Differences between the three emulators are smaller when reproducing RCP6.0 and RCP8.5 (figures available in the Supplement).

The difference between emulator approach (b) and (c) is even smaller in the other crop models than in LPJmL (figures available in the Supplement). Overall, MAD between emulated and simulated yields is up to 50% higher than LPJmL in PEGASUS, roughly twice as high in GEPIC and up to

three times as high in pDSSAT. In LPJ-GUESS, MAD between emulated and simulated yields is similar for all three emulator approaches, even though the spatial patterns of over and underestimation differ.

Using only RCP8.5 instead of all available data to train the emulators has a detrimental effect on the performance, especially for approach (a). MAD between emulated and simulated yields increases by a factor of more than three, even close to four for some GGCMs and crops, under RCP4.5. MAD for approach (b) and (c) also increases by a factor of more than two, although not as sharply as for approach (a) (figures available in the Supplement). Performance loss is lower for RCP6.0, with MAD generally less than twice as high. The emulator trained on RCP8.5 alone shows better performance in emulating RCP8.5 simulated yields than the emulator trained on all available data.

To get a more comprehensive indication of the performance of the emulator for the whole 95-year time series (instead of just the 2.5°C bin) we use all three approaches to reproduce simulated changes in crop production under

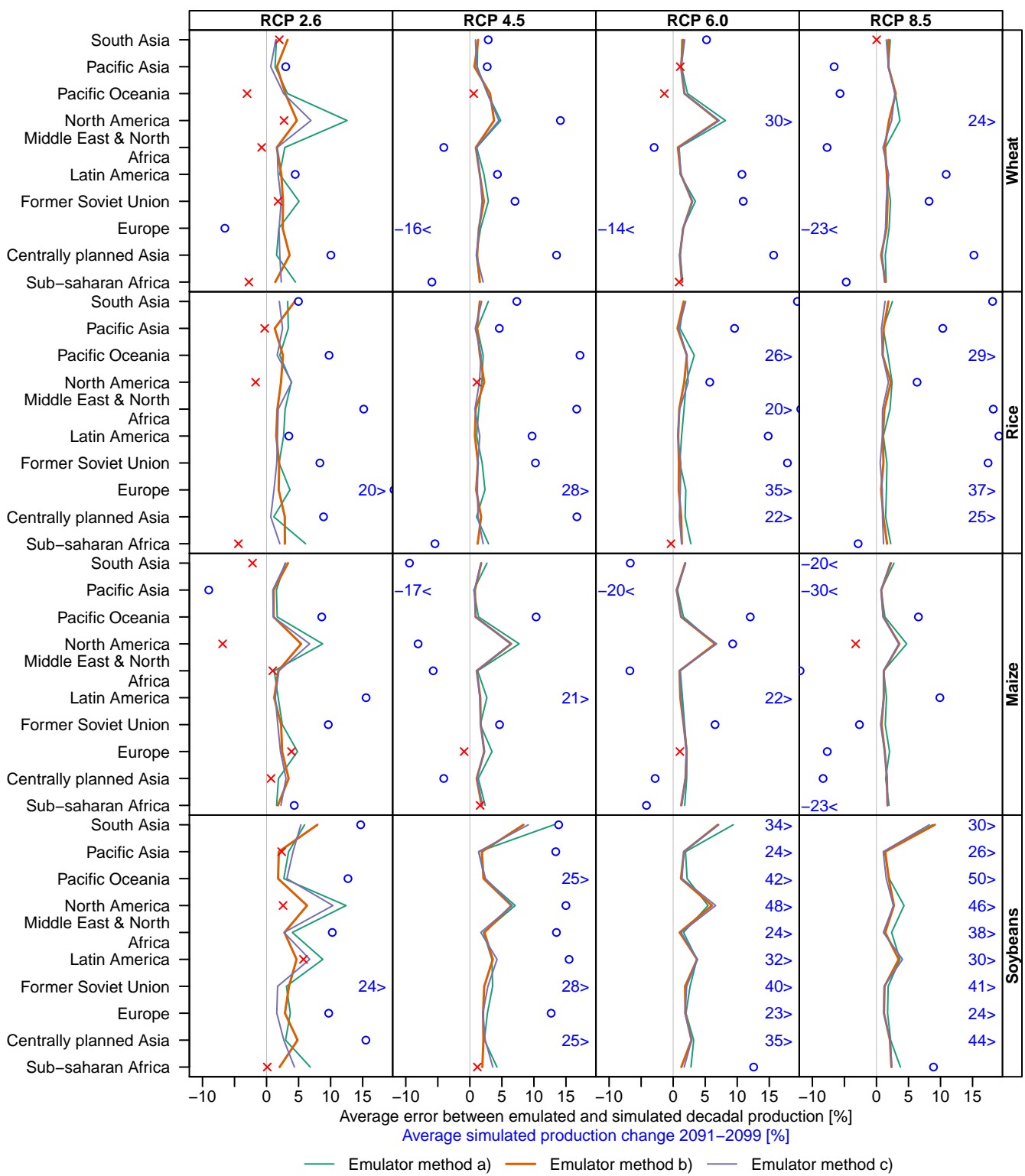

**Figure 9.** Root mean square difference (in %) between emulated and simulated regional decadal production (yields multiplied by year-2000 growing areas, combined for irrigated and rain-fed crops) for LPJmL forced by HadGEM2-ES climate projections. The emulator was built using all available data and used to reproduce yield changes in all four RCPs. For comparison, point symbols illustrate the average simulated yield change for 2091–2099 (same horizontal axis), using red crosses or blue circles depending on whether the error between emulated and simulated production is larger or smaller than the simulated change. Simulated yield changes outside the plot range are indicated by a number in the plot margin. Analogous figures for the other crop models are available in the Supplement.

RCP2.6, RCP4.5, RCP6.0, and RCP8.5, as derived for 10 large scale world regions. Grid-point yields are aggregated to the regions assuming fixed year-2000 land use and irrigation patterns. Compared to gridded yields, using production gives less weight to areas where a crop is not currently grown. Since none of the emulators is expected to capture the relatively large inter-annual variability of simulated yields we compare simulated and emulated decadal production and calculate the RMSE over all decades of the relative difference between emulated and simulated decadal production (in %) as a measure of the performance of the emulator.

Of the two approaches that estimate warming and $CO_2$-induced effects separately, approach (b) generally provides a better performance than approach (a) (see Figure 9 for LPJmL; Table 3 and the Supplement for all crop models; Figure 10 for a map of the regions). Performance of all emulator approaches varies substantially between regions. There are also considerable differences between crop models. For LPJmL, emulator approach (b) provides marginally better performance for many regions than approach (c). However, this is not consistent across the emulators for the other crop models. Taking into account that approach (b) requires additional crop model simulations with fixed $CO_2$ and that performance is mostly very similar for approach (b) and (c), the very basic interpolation approach (c) appears to provide the best compromise between emulator performance and complexity. Note though that the average difference between emulated and simulated production over the full 95-year time series is sometimes larger than the simulated production change in 2091–2099, especially in the low warming scenarios (marked by red crosses in Figure 9). Table 3 compares the RMSE between emulated and simulated crop production in the largest producing region of each crop for all five crop models.

Figure 10 illustrates the performance of emulator approach (c) in reproducing decadal maize production as simulated by LPJmL forced by HadGEM2-ES. Emulated yields generally follow the simulated trends, although large errors exist, e.g., in North America, which also stands out in Figure 9 and Figure 8. Analogous figures for all crops, emulator approaches and crop models are available in the Supplement.

Similar to the grid point results, using only RCP8.5 to train the emulators leads to a performance loss for all emulator methods and all RCPs except RCP8.5. This performance loss is larger for approach (a) than approach (b) and (c), and is generally highest for RCP4.5 (figures available in the Supplement).

## 5   Increases in Regional Crop Yield Variance

In addition to estimating the yield change associated with a rise in average temperature, it is important to consider the implications of rising variance. Climate change is expected to increase not only the average temperature, but to impact the variance of temperature and precipitation, including an increase in the frequency and duration of extreme events. For this reason, when deriving simplified relationships between yield change and global climate change, it is crucial to account not only for the mean effects of rising temperature, but also their concurrent implications for crop yield variance. Inter-annual yield variance can be computed for the same warming bins as used above for the average yields, which we do here for all four crops under the "no irrigation" scenario. The variance is calculated separately for the years of each RCP-GCM-GGCM combination falling into the 2.5°C warming bin and compared to the variance of the matching GCM-GGCM combination over the historical period (1980–2010).

The global figures show broadly similar patterns across all four crops: increases in yield variability in much of the northern hemisphere, particularly in North America, central Asia, and China; as well as in the southern mid-latitudes (Figure 11). A majority of model combinations projects decreasing variability in tropical regions (except for rice) as well as parts of Eastern Europe; but nowhere do more than 75% of the model combinations agree on a decrease in variability. In several instances increased variability occurs in highly productive regions such as in China for rice and the US, Brazil, and Argentina for soy. Wheat also has an increased variability in more than 50% of the crop model simulations over the highly productive regions in China and the U.S. Such an increase in variability, if realized, could manifest as impacts on the price, whose volatility is tightly linked to rapid changes in supply (Gilbert and Morgan, 2010).

## 6   Summary

Evaluating the impacts of climate change at different levels of global warming, and thus evaluating mitigation targets, requires a functional link between ΔGMT and regional impacts. Here we have shown that changes in crop yields, as simulated by gridded global crop models, can be reconstructed based on ΔGMT, with some limitations. The small spread of simulated yield change across the RCP scenarios as compared to the GCMs and impact models implies that projected impacts at different ΔGMT levels are not substantially dependent on the choice of emissions pathway. In this context, it has to be noted that the scenario setup of the ISIMIP crop model simulations was chosen specifically to minimize scenario-dependency by asking modellers to keep crop management fixed at present-day level or adjust it only in response to climate without any regard to the time horizons associated with adaptation or economic processes. Four models are calibrated to match present-day yield levels while LPJ-GUESS simulates potential yields assuming optimal management. Only two of the crop models allow for an adjustment of planting dates in response to climate change (GEPIC and PEGASUS, see Table 1). Three of the models

**Table 3.** Root mean square difference between emulated and simulated decadal production (expressed in % of the simulated production as in Figure 9) in the largest producing region of each crop, for all five crop models forced by HadGEM2-ES climate projections. Average across all four RCPs. The values for all combinations of models, crops, and regions, and separately for each RCP, can be found in the Supplement. Top: emulators trained on all available data; bottom: emulators trained on RCP8.5 only.

(a) Emulators trained on all available data

| Model | Wheat, Europe | | | Rice, South Asia | | | Maize, North America | | | Soybeans, Latin America | | |
|---|---|---|---|---|---|---|---|---|---|---|---|---|
| Approach | a | b | c | a | b | c | a | b | c | a | b | c |
| GEPIC | 1.334 | 1.267 | 1.215 | 3.982 | 3.037 | 2.790 | 10.099 | 9.058 | 9.360 | 3.485 | 2.550 | 2.321 |
| LPJ-GUESS | 2.242 | 2.254 | 2.213 | 4.033 | 2.163 | 3.729 | 5.870 | 5.466 | 5.359 | 2.934 | 3.025 | 2.653 |
| LPJmL | 1.777 | 1.768 | 1.596 | 2.582 | 2.371 | 1.786 | 6.923 | 5.494 | 5.846 | 4.898 | 3.870 | 4.709 |
| pDSSAT[1] | 5.363 | 3.196 | 3.550 | 7.758 | 3.606 | 4.190 | 12.218 | 6.129 | 6.149 | 3.427 | 3.662 | 3.500 |
| PEGASUS | 6.061 | 4.908 | 4.937 | n.a. | n.a. | n.a. | 8.762 | 8.533 | 8.496 | 8.762 | 8.533 | 8.496 |

(b) Emulators trained on RCP8.5 only

| Model | Wheat, Europe | | | Rice, South Asia | | | Maize, North America | | | Soybeans, Latin America | | |
|---|---|---|---|---|---|---|---|---|---|---|---|---|
| Approach | a | b | c | a | b | c | a | b | c | a | b | c |
| GEPIC | 2.159 | 1.309 | 1.396 | 6.941 | 3.541 | 3.266 | 19.091 | 9.779 | 9.664 | 5.001 | 2.654 | 2.858 |
| LPJ-GUESS | 2.579 | 2.449 | 2.486 | 5.026 | 2.656 | 4.517 | 10.034 | 7.083 | 6.866 | 3.749 | 3.355 | 2.691 |
| LPJmL | 3.814 | 2.293 | 2.415 | 4.247 | 3.040 | 2.409 | 11.954 | 5.838 | 5.950 | 5.869 | 4.607 | 5.084 |
| pDSSAT | n.a. | 4.053 | 4.392 | n.a. | 4.230 | 4.971 | n.a. | 8.290 | 7.984 | n.a. | 4.246 | 4.809 |
| PEGASUS | 8.125 | 5.167 | 5.324 | n.a. | n.a. | n.a. | 14.097 | 11.801 | 11.825 | 11.542 | 6.413 | 7.182 |

[1] Emulator approach (a) for pDSSAT only covers warming up to 3.5°C, i.e. up to 2070 under RCP8.5.

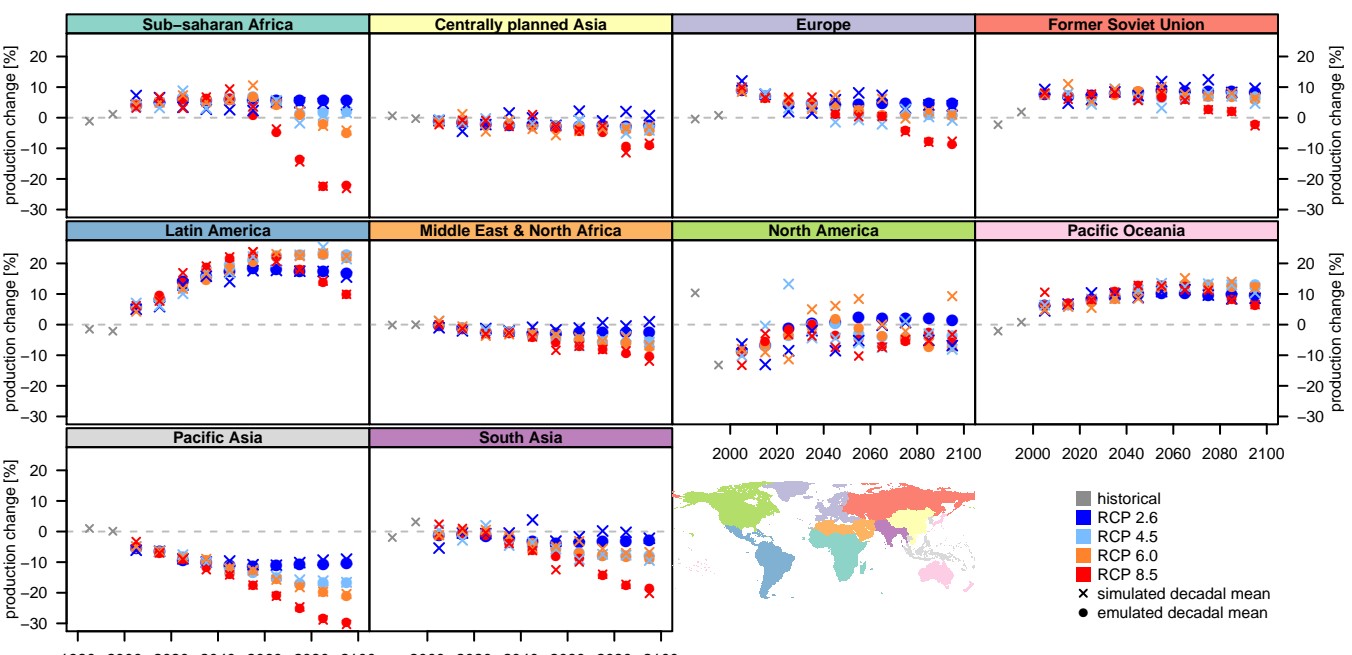

**Figure 10.** Comparison of simulated and emulated time series of regionally aggregated crop production changes for LPJmL forced by HadGEM2-ES climate projections. Results are shown for Maize and emulator approach (c). Analogous figures for the other crops, emulator approaches and GGCMs are available in the Supplement.

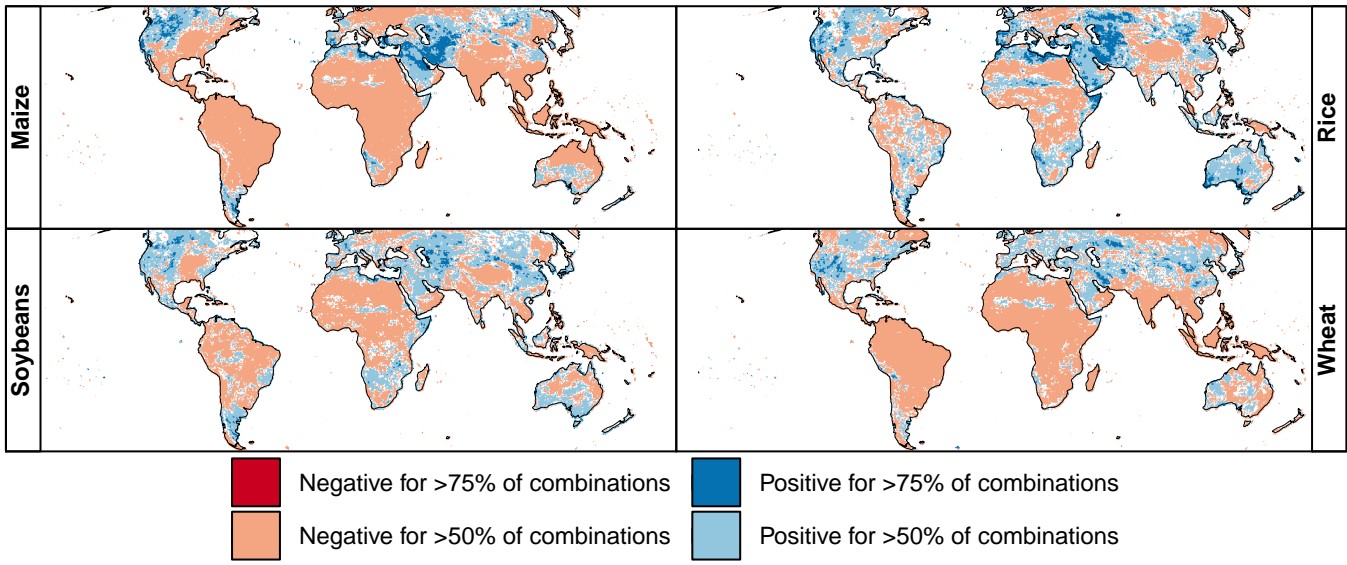

**Figure 11.** Percentage of crop model simulations (combination of a single GCM, GGCM, and RCP scenario) in the 2.5°C warming bin indicating an increase (blue) or decrease (red) in yield variance of greater than 5% compared to the historical period (1980–2010), for maize, rice, soy, and wheat under rain-fed conditions. White indicates either a less than 5% change or disagreement between the models in the direction of change. Note that only four out of five GGCMs provided results for rice. An analogous figure for irrigated conditions is available in the Supplement.

keep the total heat unit sum to reach maturity constant, assuming no change in crop cultivar which effectively leads to a shortening of the growing season. Representation of soil nutrient limitation varies substantially between models, with two models (LPJ-GUESS and LPJmL) considering no soil nutrient limitation at all, while the nutrients considered and the assumptions on fertilizer application differ between the other three models. The effects of these assumptions on yield changes simulated by the different crop models are not studied here since the focus of this study is on developing efficient emulators, but these assumptions inform both the simulated yield changes as well as the emulators which attempt to imitate the behaviour of the crop models. The results of the ISIMIP crop models have been studied in detail in Rosenzweig et al. (2014).

We have tested three different approaches for emulating crop yield change simulated by five GGCMs driven by HadGEM2-ES climate projections for four RCPs. All approaches rely on $\Delta$GMT as the main predictor of yield change at the grid scale. Two of the approaches include $pCO_2$ as an additional predictor. An approach (a) attributing the yield variation within an individual $\Delta$GMT bin of a simulation with varying $pCO_2$ solely to the change in $pCO_2$ shows the poorest overall performance. An approach (b) based on the difference between runs with and without direct $CO_2$ fertilization effects performs similarly well as a simple approach (c) using only $\Delta$GMT as a single predictor. Considering the added complexity in approach (b) compared to (c), the simple approach (c) appears in general preferable even

though it may not provide the best result in all regions. While our tests indicate that the emulators perform better for some crop models than for others we strongly advise against relying solely on results from any one particular model, but instead to always consider the full range of uncertainty spanned by the GGCMs. Similarly, different GCMs still account for more than 15% of the total variance of the ISIMIP ensemble at $\Delta$GMT=2.5°C in a number regions (Figure 4) which is why emulators should be constructed for all GCMs.

Given the availability of crop model simulations in the ISIMIP archive, emulators based on approach (a) and (c) could be constructed for all five GGCMs for the remaining four GCMs (IPSL-CM5A-LR, MIROC-ESM-CHEM, GFDL-ESM2M, NorESM1-M). Emulators based on approach (b) could only be constructed for LPJmL and pDSSAT (and PEGASUS if using only RCP8.5 for training). With its five GCMs, the ISIMIP selection essentially samples as much of the CMIP5 ensemble uncertainty as is possible with such a limited subset, but still likely underestimates the total uncertainty in future climate impacts attributable to GCMs for many regions (McSweeney and Jones, 2016). The generally good performance of approach (c) suggests that simplified predictions of large-scale agricultural yields may not require additional crop model simulations with $CO_2$ levels held at a historical level if planning to extend the GCM coverage.

While the emulators are designed to reproduce changes in average yields the impact model ensemble assembled in this study also indicates that the variability of crop yields is pro-

jected to increase in conjunction with increasing $\Delta$GMT in many important regions for the four major staple crops. Such an increase in yield volatility could have significant policy implications by affecting food prices and supplies, although management assumptions as well as model-structural limitations of the GGCMs to account for crop stress factors may impact the models' ability to project future changes in variability.

The scalability of mean yields is conducive to the development of predictor functions relating $\Delta$GMT, or other aggregate climate variables readily available from simplified climate models (such as $pCO_2$) to regional or global mean crop yield impacts. This lays the groundwork for a further exploration of the economic impacts of climate change encountered at target warming levels or over policy relevant regions.

**Data availability.** The coefficients estimated with Equations 1 to 3 are available as a Supplement, along with supplementary figures and RMSE estimates, at https://doi.org/10.5281/zenodo.1194045. The GGCM simulations that the analysis in this paper is based on are available through https://esg.pik-potsdam.de/search/isimip-ft/, with additional documentation available on the ISIMIP website https://www.isimip.org/outputdata/caveats-fast-track/

**Acknowledgements.** This work was supported within the framework of the Leibniz Competition (SAW-2013 PIK-5), by the EU FP7 HELIX project (grant no. 603864), and by the German Federal Ministry for the Environment, Nature Conservation and Nuclear Safety (16_II_148_Global_A_IMPACT).

For their roles in producing, coordinating, and making available the ISIMIP model output, we acknowledge the modelling groups (listed in Table 1 of this paper) and the ISIMIP cross sectoral science team.

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
