# Peer review of "Changes in crop yields and their variability at different levels of global warming"

_Earth System Dynamics, 2017_

## Referee Comment (RC1) · Anonymous Referee #1 · 26 Sep 2017

General comments:

The manuscript "Changes in crop yields and their variability at different levels of global warming" tests an emulator approach to estimate crop yield impacts for arbitrary CO2 emission scenarios based on ISIMIP multi-impact model projections. From a topical point of view, I think this study falls within the general scope of Earth System Dynamics journal. Clearly, the novelty of this paper lies in designing statistical methods to emulate crop yield impacts described in terms of dGMT, going beyond the existing emission scenarios included in the ISIMIP data-cube. Such emulator could provide a shortcut for fast assessment of yield impacts for a range of mitigation targets, thus extending the capacity of ISIMIP ensemble. The authors demonstrate that both mean yields and yield variability are, to a certain degree, directly scalable across emission scenarios, without

additional explicit RCP-GCM-GGCM simulations. I believe that the ISIMIP data cube alone, as described by the authors, already provides a powerful tool for linking crop yield changes with arbitrary dGMTs, but has limitations with respect to arbitrary $CO_2$ levels. This is definitely a valuable work, with clear and sound conclusions, and after a moderate revision it would be an original contribution to community of crop impact modellers and for Integrated Assessment Models in general.

Overall, the manuscript is of high technical standard. However, I must say that I had problems with following the methods at some places since various methodical aspects are scattered across the whole document, which makes data reproducibility more difficult. The main "weakness" comes with using the linear-regression (or weighted-average) emulators at a grid basis without further testing its significance. Since the emulators for individual dGMT are based on intercept ($a_0$) and regression slope ($a_1$) of the fitted linear models (determining climate- and emission-induced yield changes, respectively), it would be useful to add a step of testing whether or not these values and the fit overall are statistically significant, and exclude those grids for which it is not so. One could expect that yield response along a narrow range of $CO_2$ increase (such as for a low dGMT) may be quite noisy, and not necessarily captured by linear models in a significant way. This might be the case for some GGCMs and regions where crop yield changes in a response to shifts in T and $CO_2$ are limited by other constrains, such as nutrient deficiency for example.

Specific comments:

P3, paragraph 115: soya is not a cereal crop

P4, p.150: "...generated for four RCPs..." which RCP scenarios were used here? It would be nice to list the emission scenarios here as this is the first time the scenarios are mentioned in the Methods section.

P4, p.165: "...were forced by climate change projections from HadGEM2-ES, RCP8.5..." I wonder if this statement is correct. The authors refer to "YnoCO2" also

for other RCPs in Figure 5, if I am not mistaken.

P5, Table 1: I agree that the model-related assumptions have to be short, but some statements in Table 1 are jargon difficult for others to understand. For example, for LPJ-GUESS, is the thermal time needed for maturity adjusted over time? It is not clear from the explanation in Table 1...if so, there is probably no shortening of the growing season effect in this model, which may explain more positive effects of warming on wheat in Figure 2, in contrast to other models. Next, what does "decadal adjustment of winter and spring wheat sowing areas based on temperature" mean? Then, do LPJ-GUESS and LPJmL use no information on fertilization, meaning there is no limitation by nutrients at all? If the same assumption applies for both LPJ-GUESS and LPJmL, why not using the same description for both? Please, use harmonized and clearer descriptions that would allow readers to understand fundamental differences in the models, and which implications it might bring. Overall, I understand that the differences in yield projections due to unique specifications of individual GGCMs are not topic in this manuscript, but it would be very useful to discuss some results with a deeper insight into individual models where appropriate (such as that some models simulate nutrient-unlimited yields while the others not).

p.7: It is obvious from Fig 1 that different dGMT bins would bring different number of years into analysis (1) and (2). Time intervals for some dGMT bins may be quite short (e.g. 10 years), and probably not sufficiently long to smooth possible short-term inter-annual fluctuations and anomalies in simulated yields. In other words, the mean yield change could be very sensitive to anomalous years. A concept of climatic normal is usually used to eliminate anomalous fluctuations. I am not sure about the implications for this approach, but the authors should consider and possibly discuss this aspect.

p.12, Figure 5: please add information whether the linear regressions are statistically significant. Free scaling of y-axis may work better here to see the data scatter more clearly.

p. 16, Fig. 7: as discussed in the manuscript, negative impacts of $CO_2$ are contraintuitive, even though they may make sense from the statistical point of view. Maybe a rule that all negative impacts of increased $CO_2$ are reverted to zero would make more sense.

P17. P. 405 and elsewhere: "...change on potential yields..." This statement here (and also elsewhere) is confusing for me. Do the authors estimate yield potentials or yields under historic management?

P17.420: (ao(dGMT)) should be (dYCLIM(dGMT))

Section 5, p.23: In my opinion, de-trending of simulated yields within a dGMT bin by RCP-specific yield averages provides only an artificial variability, since it combines different RCP projections together. I am not an expert in climate modelling, but I assume that each radiative forcing scenario in a GCM generates a unique temporal variability of meteorological variables, and I have my doubts that mixing RPC-specific simulations together makes sense in terms of dGMT-specific variability. I am not sure that I understand the concept in Section 5 correctly though.

p. 25: "......implies that projected impacts at different dGMT levels are not substantially dependent on the choice of emissions pathways." This might be true unless lead time is considered. Time horizons associated with individual dGMT levels are of high importance for adaptation and economic processes.

---

## Editor Comment (EC1) · V. Arora (Editor) · 19 Dec 2017

Authors evaluate the potential of a simple modelling framework to reproduce the geographical distribution of changes in crop yield based on solely the change the global mean temperature ($\Delta$GMT) and change in atmospheric $CO_2$ concentration. This simple modelling framework is calibrated based on results from five spatially explicit process-based crop models that are driven with data from five Earth system model (ESM) for four future RCP scenarios. The potential uses of such a tool are obvious for impact studies but in its current form the manuscript is somewhat difficult to follow. As a reader, I felt several clarifications are needed. In particular, I wasn't able to completely follow the three emulator models used to reproduce the results from the full crop models and still unclear how the different temperature bins come into picture. In addition,

it would be helpful for a reader if the plots and graphics were bit more readable as suggested in comments below.

I also have comments marked on an annotated version of the manuscript which I attach as a supplement. Most comments from my hand written notes are summarized below except the minor comments for which I request you to see the attached annotated version of your manuscript. Hope my hand writing is easily readable.

Specific comments

Page 3, lines 100-105. I find it difficult to believe that spatially explicit crops models (driven with spatially explicit climate information even though it may require some kind of scaling) will be worse off than the approach presented in this manuscript.

Page 4, lines 133-135. "The simulated impacts of climate and CO2 changes on global and regional crop yields are shown to be related to global mean temperature change, and to be largely independent of the emissions scenario". There is a bit of circular argument here and then later on in Figure 4 where most of the variance in crop yield is explained by crop models. I think, by choosing a specified $\Delta$GMT you have made $\Delta$GMT independent of emission scenarios and ESMs. What remains scenario and ESM dependent is the year when this specified $\Delta$GMT is reached. Results from climate models show that the typical spatial pattern of temperature change is also similar across ESMs – that is higher warming over land than over ocean and higher warming at high latitudes than in tropics. Combined with specified $\Delta$GMT this means that it is somewhat expected that most of the variance in crop yield will be explained by crop models.

Section 2. page 4.) Please 1) specify the time period of projections, 2) introduce the four RCPs and what they imply (i.e. low, medium and high emissions scenarios), 3) introduce CMIP5 a bit more and 4) provide 1 or 2 sentence about how bias correction is done.

Table 1. page 5. The description of "Fertilizer use" needs to be made more consistent across the five crop models. For example, for LPJ-GUESS I am not sure what does "no consideration of spatial and temporal changes in nutrient limitation" means. Does this mean nutrient constraints are not considered or a specified fertilizer application rate is assumed for all times and all parts of the world.

Page 7, lines 200-221. What is MIRCA 2000? Is this an observation-based product?

Figure 3 and for other similar figures. Please consider putting the figure titles (Maize, Rice, Soybeans and Wheat) in horizontal format and a bigger font size as suggested in the annotated manuscript. Also, as a reader I was wondering what are the limits on the horizontal colour bar. I found the colour bar a bit difficult to interpret. If the white colour represents yield change of less than $\pm$ 5% then how can light blue and light red colours represent changes of more than 50%. For example, +6% will likely be indicated by light blue colour but this is not more than 50%.

Page 12, lines 333-334, "To quantify the extent of the $CO_2$ induced scenario dependence and its potential reduction at each grid point . . .". I am unable to understand what do "$CO_2$ induced scenario dependence" and "potential reduction" refer to. There are two possible aspects here and I am not sure which one is right in his context. The first is that of $CO_2$ fertilization – is this what is being referred to in this sentence. Second, note that just like by choosing a specified $\Delta$GMT of 2.5 degree Celsius the dependence on emissions scenario and ESM has been reduced $\Delta$GMT is also related to change in atmospheric $CO_2$ concentration. Is this what is being referred to here? Page 13. Equations (1) and (2) are the crux of the paper. Yet, I unable to understand how these equations are used. Perhaps if more equations were used to describe each and every term of these equations it would have been easier to follow them. For example, the change in yield is actually a two-dimensional quantity (i.e. it depends on the geographical location) at a given time (represented by i in equations 1 and 2). Perhaps if (t) can be used to represent time and not as subscript and if these equations were written again properly it would be easier to understand the objective of these equations. I am also unable to appreciate if equation (1) and (2) were applied at each individual grid cell or to the time series of globally-summed yield. In this context, I am also unable to understand what do the different temperature bins refer to.

In Figures 6 and 7, the absolute changes in yield are small (since they are mostly white in colour) yet the percentage changes are huge because those percentage changes are corresponding to small absolute values (as mentioned in the manuscript). Perhaps if the percentage changes can be masked over regions of low crop yields then these figures will be much more easier to interpret than the current percentage change figures. Alternatively, maybe the colour scale for figures with absolute yield changes can be changed and the percentage change figures can be removed.

Page 18, lines 439-442. "While approach (b) requires a pair of crop model simulations – one with time-varying pCO2 and one with fixed pCO2, approach (a) only requires the default simulations with time-varying pCO2". I wasn't able to appreciate this. So perhaps this should be mentioned earlier on where the three emulator approaches are presented.

Page 20, line 485-486. While shown in Lotze-Campen et al., 2008 please include the map of the 10 world regions in this manuscript as well.

Page 20, line 486-487. "Compared to potential yields, using production gives less weight . . .". Please introduce (if necessary) and differentiate yield from production. Not all readers can be expected to appreciate this difference between the two.

Figure 10. It is very hard to see the thin lines corresponding to each RCP. Please consider including thick lines for say 8 or 10 year moving average values which would likely yield a more fair comparison with values from the emulators which do not account for the effect of climate variability on crop yield. Also, please consider using a different set of colours for the four RCPs. The green and the blue seem very similar and the orange and red are pretty close colours.

Page 23, lines 541-544. I am unable to understand this. Also, I am unsure how there can be any variance attributed to $CO_2$ effects. If this refers to the $CO_2$ fertilization effect – doesn't $CO_2$ changes gradually in all RCP scenarios. So yes, while there is a trend in specified $CO_2$, there is not year-to-year variability. Or, am I misinterpreting this.

I think, it should be made clear in the manuscript that the emulators do not capture the year-to-year variability but rather the long term trend in crop yield.

Please also note the supplement to this comment:
https://www.earth-syst-dynam-discuss.net/esd-2017-69/esd-2017-69-EC1-supplement.pdf

**Supplement:**

[revised manuscript text omitted]

*This was expected since you chose a temperature change value of 2.5°C ( Had you chosen a specified year climate model & scenario would contribute to variance as well.*

The simulated yield values at each grid point and within each GMT bin are subject to variation due
to the selection of impact model, GCM forcing, and emissions scenario. When considering all of
these factors, the variance attributable to the impact model selection is much greater than that
associated with the GCM or scenario choice in most regions (Figure 4). This holds for rainfed as
well as irrigated simulations and at all global mean warming bins above 1°C. The predominance of
the impact model component in total variance is particularly evident in the middle to high
latitudes for all four cereal crops, where impact model variance accounts for up to 90% of the grid
point variance at 2.5°C.

[Figure]

**Figure 4.** Fraction of total variance attributable to the impact models (GGCMs, left), climate
models (GCMs, middle), and scenarios (RCPs, right) for each crop. Figure shown for rain-fed runs
at ΔGMT=2.5°C warming; an analogous figure for irrigated runs is provided as supplementary
online material.

**3.2 Direct impacts of increasing pCO2**

In addition to air temperature warming, pCO2 has a direct influence on crop yields. As it varies
within the different ΔGMT bins, it is expected to induce part of the fluctuations of the yield
changes at given GMT levels. We find that this CO2 effect is not scenario dependent (see Figure 5
for the global average effect within the LPJmL simulations), consistent with a short response time
of plants to pCO2 changes.

[Figure]

[Figure]

**Figure 5.** Difference in global mean yield change (sum of rainfed and irrigated, and weigthed by present-day growing areas) between the default ($Y_{CO2}$) and fixed CO2 simulations ($Y_{noCO2}$), for each crop over the range of pCO2 associated with the $\Delta$GMT =2.5°C bin. Results are as simulated by LPJmL forced with output from HadGEM2-ES. Each color represents an emission scenario and black dotted lines indicate the linear best fit for each crop.

*→ from what?*

As expected, the differences increase with heightened atmospheric CO2 level under all emissions scenarios, implying a stronger CO2 fertilization impact with increased pCO2. A least squares fit to the yield differences versus greenhouse gas level within each $\Delta$GMT bin allows for a quantification of the direct CO2 effect at each level of warming based on global pCO2, rather than the emissions pathway. The underlying assumption is that the effect of the temperature variation within the 0.5°C range of each $\Delta$GMT bin will be minimal compared to the effect of the CO2 variation across all RCPs.

*not sure what is being implied here.*

To quantify the extent of the CO2 induced scenario dependence and its potential reduction at each grid point, we use two methods to determine the CO2 effect on crop yields within each global mean temperature bin:

*what does this reduction refers to? Doesn't CO2 fertilization enhance yield?*

[Figure]

*Can u pls remind the reader again what is this period? Is it 1980-2010?*

*Are these two formulations a fit to explain the data in Fig 5.*

(a) By linear regression of absolute yield changes with respect to the historical reference period ($\Delta Y_{CO2}$) on $CO_2$ concentration within the individual global mean warming bins, i.e. by fitting the following model

$$\Delta Y_{CO2,\,i} = \Delta Y_{CLIM} + a_1{}^* (pCO_{2,\,i} - 370 \text{ ppm}) + \varepsilon_i, \qquad (1)$$

where i indicates the individual year within the relevant $\Delta GMT$ bin, and $\varepsilon_i \sim N(0, \sigma^2)$ represents the residual error. The statistical model allows for the estimation of the purely climate-induced yield change $\Delta Y_{CLIM}$ at a fixed year-2000 concentration of CO2 of 370 ppm.

(b) By linear regression of the within-bin differences between the default crop simulations ($Y_{CO2}$) and the fixed CO2 run ($Y_{noCO2}$) on the underlying CO2 concentration in the default simulation:

$$(Y_{CO2,\,i} - Y_{noCO2,\,i}) = a_0 + a_1{}^* (pCO_{2,i} - 370 \text{ ppm}) + \varepsilon_i, \qquad (2)$$

where i indicates the individual year and $\varepsilon_i \sim N(0,\,^2)$ represents the residual error. In this case the purely climate-induced yield change $\Delta Y_{CLIM}(\Delta GMT)$ is given by the yield change in the fixed CO2 run, $\Delta Y_{noCO2}(\Delta GMT)$, and an additive correction $a_0$. This correction accounts for the different levels of pCO2 in the fixed-CO2 run across different models; it is zero if the pCO2 in the fixed-CO2 run is 370 ppm.

*If historical reference period is 1980-2010, doesn't this mean $\Delta Y_{CLIM}$ is around zero.*

*What is $\Delta Y_{CLIM}$? Can u pls show an eqn?*

*Do you derive $a_0$ & $a_1$ for each crop type (wheat, rice, maize and soybean)?*

[Figure]

*(handwritten annotations)*
hard to read
These two are not similar. Approach b) gives much smaller yield changes than a).
↳ increase font size

**Figure 6. Climate change-induced yield changes at ΔGMT= 2.5°C of global warming and year 2000 pCO2 level (370ppm).** Left column: Patterns of $\Delta Y_{CLIM}$ derived at each grid point by method (a) (see equation (1)). Right column: Patterns of $\Delta Y_{noCO2}(2.5°C)+ a_0$, derived by method (b) (see equation (2)). Both types of patterns are derived from LPJmL simulations forced by HadGEM2-ES assuming rain-fed conditions and are expressed in percentage of change relative to the historical average yield at each grid point. Rows: Different crop types. Top panel shows relative differences,

[Figure]

bottom panel shows absolute differences. Analogous figures for irrigated conditions and for different GGCMs are available as supplementary online material.

→ why do you keep referring to temperature bins if all the analysis is focussed for $\Delta GMT = 2.5 \pm 0.25\ °C$.

→ I understand $Y_{no CO_2}$ is the yield from simulations with no $CO_2$ fertilization but what is $\Delta Y_{no CO_2}$ — it's difference compared to what?

→ what is the purpose of eqns 1 and 2. Can't the effect of climate and $CO_2$ be seen by analyzing results from simulations with and without $CO_2$ fertilization effect directly without the use of eqns 1 and 2.

[Figure]

[Figure]

**Figure 7. CO$_2$-induced yield changes at 2.5°C of global warming.** Analogous to Fig. 6 but for the bin-specific CO$_2$ scaling coefficients a$_1$. **Rows:** Different crop types. Top panel shows relative differences, bottom panel shows absolute differences. Analogous figures for irrigated conditions and for different GGCMs are available as supplementary online material. → *see note on Fig 6*

The two methods result in broadly similar patterns for the climate change-induced relative yield changes (i.e., excluding direct CO2 fertilization effects), with yield increases in the high latitudes

and decreases in the tropics and subtropics, broadly speaking (Fig. 6). However, the magnitudes of the changes are much larger with method (a) (Fig. 6, lower panel). Some regional differences also occur between the two methods, such as for rice where there is disagreement on the direction of yield changes in southeast Asia.

*Perhaps you should mask out regions with yield less than some threshold.*

In relative terms (estimated climate change-induced yield change divided by simulated present-day yield), both methods show very large values of frequently alternating sign in areas such as the Arabian peninsula or the northern Sahel (Fig. 6, upper panel). This is likely due to the very low present-day yield potential in these regions, leading to division by values close to zero. In the regional evaluation of the different emulator methods below, we will account for these regional
differences in baseline yields by weighting potential yield changes by present-day growing areas.

*The upper panels (% change) of Figs 6 & 7 are very confusing to look at.*

The estimates of CO2-induced yield changes also differ between the two methods (Figure 7). Method (b) results in a positive CO2 effect in most regions, except for some low-yielding areas and the potentially important cases of soybean in southern and eastern South America, and rice in
north-west India and Pakistan, where it results in a negative effect of rising pCO2 on yield. With method (a) on the other hand, areas of negative estimated CO2 effect are much more widespread, and generally the magnitudes of the estimated CO2 effect are again much larger than with method (b). As a preliminary conclusion, the results obtained with method (b) for the separate effects of climate change and pCO2 change on potential yields appear more realistic than those
obtained with method (a).

*In the absence of an understanding of the purpose of approaches a) and b), I am asking myself why can't the approaches be evaluated against the simulated climate & CO2 effects from the models using results from simulations with and without the CO2 fertilization effect.*

**4. Validation of three emulator approaches**

*still struggling with this*

Based on the climate-induced patterns (assuming fixed year 2000 levels of CO2) of relative yield
changes and the associated within-bin relationship between CO2 and crop yields identified in section 3, we propose the following two-step interpolation method to compute crop yield changes for any given pair of $\Delta GMT$ and pCO2, using either of the above regression methods (a) or (b):

[revised manuscript text omitted]

*Hard to understand without equation.*

*I think, it should be made clear that the approach presented in This paper doesn't emulate variability. It only attempts to emulate mean or rather long term trend.*

[Figure]

[Figure]

**570  6. Summary**

Evaluating the impacts of climate change at different levels of global warming, and thus evaluating mitigation targets, requires a functional link between ΔGMT and regional impacts. Here we have shown that changes in crop yields, as simulated by gridded global crop models, can be reconstructed based on ΔGMT, with some limitations. The small spread of simulated yield change across the RCP scenarios as compared to the GCMs and impact models implies that projected impacts at different ΔGMT levels are not substantially dependent on the choice of emissions pathway.

We have tested three different approaches for emulating crop yield changes simulated by GGCMs, two of which include pCO2 as an additional predictor. An approach (a) attributing the variation within an individual ΔGMT bin of a simulation with varying pCO2 solely to the change in pCO2 shows the poorest overall performance. An approach (b) based on the difference between runs with and without direct CO2 fertilization effects performs similarly well as a simple approach (c)

[revised manuscript text omitted]

---

## Referee Comment (RC2) · Anonymous Referee #2 · 18 Jan 2018

This paper succeeds in showing the relative contributions to the spread of crop yield projections from impact models, climate models and scenarios. Further, it also succeeds in demonstrating that there is potential utility to functions relating change in GMT to yield impacts in the future. Clear and precise outlining of methods and data availability makes the results of this paper easily traceable and its contents easily replicable by fellow scientists.

This paper would benefit from further discussion of methodological limitations. In particular, the following points should be addressed:

i) 5 GCMs are used to obtain climate projections. The authors should discuss the representativeness of these 5 GCMs with regards to the CMIP 5 ensemble.

ii) Section 5 describes projected increases in regional crop yield variance. This section should also include a discussion of the extent of uncertainty present in these projections of rising variance.

The summary section should incorporate discussion of the above two points in relation to the strength of the conclusions drawn.

In addition to elaborating on methodological limitations, the following points require clarification and elaboration respectively:

Lines 393 – 394: The use of the word "likely" needs to be clarified here. Please provide a definitive answer as to whether or not very low present-day yield potential in these regions is leading to division by values close to zero.

Lines 396 – 405: Please give an explanation, or hypothesis for the negative effects of CO2 in the two "potentially important" regions mentioned.

In terms of mathematical formulae, symbols, abbreviations and units, the following should be addressed:

Lines 169 onwards: Please use a clearer term for fixed CO2 than YnoCO2 .

Lines 340 and 350: Please describe what a1 represents in each equation in word form.

Line 434: Please correct the use of <> in this equation.

In terms of changes to figures, the following should be addressed:

Figures 2,4,6,7 and 8: All of these figures need to be much larger to increase their readability. If possible, each figure should be on its own page.

Figure 5: This figure is only for the LPJML model, please explain the rationale for model selection or point to where other model results can be found.

[Figure]

---

## Author Comment (AC1) · 15 Feb 2018

**Response to**

**Anonymous Referee #1**

---The referee's comments are quoted in black, and our responses in *blue italics*.---

General comments:

The manuscript "Changes in crop yields and their variability at different levels of global warming" tests an emulator approach to estimate crop yield impacts for arbitrary $CO_2$ emission scenarios based on ISIMIP multi-impact model projections. From a topical point of view, I think this study falls within the general scope of Earth System Dynamics journal. Clearly, the novelty of this paper lies in designing statistical methods to emulate crop yield impacts described in terms of dGMT, going beyond the existing emission scenarios included in the ISIMIP data-cube. Such emulator could provide a shortcut for fast assessment of yield impacts for a range of mitigation targets, thus extending the capacity of ISIMIP ensemble. The authors demonstrate that both mean yields and yield variability are, to a certain degree, directly scalable across emission scenarios, without additional explicit RCP-GCM-GGCM simulations. I believe that the ISIMIP data cube alone, as described by the authors, already provides a powerful tool for linking crop yield changes with arbitrary dGMTs, but has limitations with respect to arbitrary $CO_2$ levels. This is definitely a valuable work, with clear and sound conclusions, and after a moderate revision it would be an original contribution to community of crop impact modellers and for Integrated Assessment Models in general.

Overall, the manuscript is of high technical standard. However, I must say that I had problems with following the methods at some places since various methodical aspects are scattered across the whole document, which makes data reproducibility more difficult. The main "weakness" comes with using the linear-regression (or weighted-average) emulators at a grid basis without further testing its significance. Since the emulators for individual dGMT are based on intercept ($a0$) and regression slope ($a1$) of the fitted linear models (determining climate- and emission-induced yield changes, respectively), it would be useful to add a step of testing whether or not these values and the fit overall are statistically significant, and exclude those grids for which it is not so. One could expect that yield response along a narrow range of $CO_2$ increase (such as for a low dGMT) may be quite noisy, and not necessarily captured by linear models in a significant way. This might be the case for some GGCMs and regions where crop yield changes in a response to shifts in T and $CO_2$ are limited by other constrains, such as nutrient deficiency for example.

*Reply: We have added a test whether the derived fits are statistically significant. We have also changed the regression for method b) (equation 2) slightly, which increases the number of grid-cells with significant fits. We will add language to the paper to discuss the implications of limited (and noisy) training data for the emulator.*

Specific comments:

P3, paragraph 115: soya is not a cereal crop

*Reply: Thanks for pointing this out. We will rephrase it.*

P4, p.150: "...generated for four RCPs..." which RCP scenarios were used here? It would be nice to list the emission scenarios here as this is the first time the scenarios are mentioned in the Methods section.

*Reply: We will add the RCPs used (which are all RCPs considered in CMIP5).*

P4, p.165: "...were forced by climate change projections from HadGEM2-ES, RCP8.5..." I wonder if this statement is correct. The authors refer to "YnoCO2" also for other RCPs in Figure 5, if I am not mistaken.

*Reply: Only fixed CO2 simulations (YnoCO2) driven by HadGEM2-ES RCP8.5 were used for emulator approach b) (linear regression according to equation 2). However, some crop models such as LPJmL also provided fixed CO2 simulations for other RCPs. We will clarify this.*

P5, Table 1: I agree that the model-related assumptions have to be short, but some statements in Table 1 are jargon difficult for others to understand. For example, for LPJ-GUESS, is the thermal time needed for maturity adjusted over time? It is not clear from the explanation in Table 1 … if so, there is probably no shortening of the growing season effect in this model, which may explain more positive effects of warming on wheat in Figure 2, in contrast to other models. Next, what does "decadal adjustment of winter and spring wheat sowing areas based on temperature" mean? Then, do LPJ-GUESS and LPJmL use no information on fertilization, meaning there is no limitation by nutrients at all? If the same assumption applies for both LPJ-GUESS and LPJmL, why not using the same description for both? Please, use harmonized and clearer descriptions that would allow readers to understand fundamental differences in the models, and which implications it might bring. Overall, I understand that the differences in yield projections due to unique specifications of individual GGCMs are not topic in this manuscript, but it would be very useful to discuss some results with a deeper insight into individual models where appropriate (such as that some models simulate nutrient-unlimited yields while the others not).

*Reply: Regarding your first question: In LPJ-GUESS, the thermal time needed for maturity is adjusted over time to keep the growing season length constant. This is indeed different from most of the other crop models (except PEGASUS) which keep the heat sum constant, leading to a shorter growing season length under higher temperatures. Regarding your second question: The decision to grow wheat as either winter or spring wheat in any particular grid-cell and year is left to the models. In GEPIC, this can change over time, compared to other crop models which use fixed sowing dates. Regarding the third question: Neither LPJ-GUESS nor LPJmL consider nutrient limitation of plant growth.*
*After further consideration, we have decided that much of the information in Table 1 is not directly relevant for our study. Our focus is on developing and comparing three emulator methods to approximate yield changes simulated by the crop models; rather than on the differences between the crop models themselves. These have been studied elsewhere for the same ISIMIP crop models (e.g. Rosenzweig et al., 2014). Therefore, we will remove parts of this table and refer to the appropriate literature for readers interested in the crop model details; the information remaining in table 1 will be clarified.*

p.7: It is obvious from Fig 1 that different dGMT bins would bring different number of years into analysis (1) and (2). Time intervals for some dGMT bins may be quite short (e.g. 10 years), and probably not sufficiently long to smooth possible short-term inter-annual fluctuations and anomalies in simulated yields. In other words, the mean yield change could be very sensitive to anomalous years. A concept of climatic normal is usually used to eliminate anomalous fluctuations. I am not sure about the implications for this approach, but the authors should consider and possibly discuss this aspect.

*Reply: Climatic normals are usually based on 30-year time slices which, especially in the case of RCP8.5, can encompass a significant range of warming (e.g. more than 1.5°C of warming 2071–2100). In comparison, most 0.5°C-wide dGMT bins are much shorter. Since dGMT bins are independent of the emissions scenario this offers an opportunity to combine data from different scenarios and extend the number of years of data in a bin. For example, a total of 66 years from four RCPs fall into the 1°C warming bin. Combining data from different RCPs is not possible for high levels of warming only reached by RCP8.5. We note, however, that all bins*

*used for training in our study contain at least 7 years, which does allow for interpolation,
although potentially with large noise. To clarify this, we will test for statistical significance of
derived fits and discuss implications of limited training data for the emulator.*

p.12, Figure 5: please add information whether the linear regressions are statistically
significant. Free scaling of y-axis may work better here to see the data scatter more clearly.

*Reply: The intention in Fig. 5 is to show that the effect of pCO2 within a given dGMT bin does
not depend on the scenario. Figure 5 illustrates this by showing the globally averaged yield
difference between the default and the fixed-$CO_2$ simulations. The linear regression line was
added as a visual aid, but is not actually used for any further analysis. We realize that it would
probably be more helpful to show separate regression lines for the different RCPs to
underscore the argument. We will add these in the revision, and will also use free y-axis
scaling. A test of statistical significance is not necessary in our opinion since these regressions
are solely for illustration; but we will consider showing confidence intervals along with the
regression lines.*

p. 16, Fig. 7: as discussed in the manuscript, negative impacts of CO2 are contra-intuitive,
even though they may make sense from the statistical point of view. Maybe a rule that all
negative impacts of increased CO2 are reverted to zero would make more sense.

*Reply: For emulator approach a) (equation 1) both the temperature effect and the CO2 effect
are derived from the same regression. In this case, it's not possible to set negative CO2
effects to zero because both effects are linked. For emulator approach b) (equation 2) only the
CO2 effect is derived from the regression. Since submitting the first version of the paper, we
have changed the regression slightly to derive the correction term a0 in equation 2 more
robustly. (We have explicitly accounted for between-model differences in the baseline CO2
value, whereas earlier these were only implicitly included in the estimation of a0.) This has
eliminated any negative CO2 effects. We will update the paper with the revised method.*

P17. P. 405 and elsewhere: "...change on potential yields..." This statement here (and also
elsewhere) is confusing for me. Do the authors estimate yield potentials or yields under
historic management?

*Reply: All GGCMs except LPJ-GUESS simulate yields assuming present-day management as
a starting point. In this case, "potential yields" refers to the fact the yield simulations are not
only carried out for areas where the crops are grown today. Instead, yields projections cover
all areas where the crop can grow. We will try to clarify this more.*

P17.420: (ao(dGMT)) should be (dYCLIM(dGMT))

*Reply: Thank you for pointing out this inconsistency with equation 1. We will change it in the
revision.*

Section 5, p.23: In my opinion, de-trending of simulated yields within a dGMT bin by RCP-
specific yield averages provides only an artificial variability, since it combines different RCP
projections together. I am not an expert in climate modelling, but I assume that each radiative
forcing scenario in a GCM generates a unique temporal variability of meteorological variables,
and I have my doubts that mixing RPC-specific simulations together makes sense in terms of
dGMT-specific variability. I am not sure that I understand the concept in Section 5 correctly
though.

*Reply: We do not combine results from different RCP projections. Changes in variance are
computed for each RCP-GCM-GGCM combination separately. Figure 11 shows the*

*percentage of RCP-GCM-GGCM combinations that experience an increase or decrease of yield variance of greater than 5% at 2.5°C warming.*

p. 25: "......implies that projected impacts at different dGMT levels are not substantially dependent on the choice of emissions pathways." This might be true unless lead time is considered. Time horizons associated with individual dGMT levels are of high importance for adaptation and economic processes.

*Reply: Time horizons associated with adaptation and economic processes are not considered in the ISIMIP crop simulations. Assumptions on management in the GGCMs are either constant over time or are driven solely by climate.*

---

## Author Comment (AC2) · 15 Feb 2018

**Response to**

**V. Arora (Editor)**

---The editor's review comments are quoted in black, and our responses in *blue italics*.---

Authors evaluate the potential of a simple modelling framework to reproduce the geographical distribution of changes in crop yield based on solely the change the global mean temperature (ΔGMT) and change in atmospheric CO2 concentration. This simple modelling framework is calibrated based on results from five spatially explicit process-based crop models that are driven with data from five Earth system model (ESM) for four future RCP scenarios. The potential uses of such a tool are obvious for impact studies but in its current form the manuscript is somewhat difficult to follow. As a reader, I felt several clarifications are needed. In particular, I wasn't able to completely follow the three emulator models used to reproduce the results from the full crop models and still unclear how the different temperature bins come into picture. In addition,it would be helpful for a reader if the plots and graphics were bit more readable as suggested in comments below.
I also have comments marked on an annotated version of the manuscript which I attach as a supplement. Most comments from my hand written notes are summarized below except the minor comments for which I request you to see the attached annotated version of your manuscript. Hope my hand writing is easily readable.

*Reply: Regarding the temperature bins: We expect that yield changes may not necessarily scale linearly with temperature over a large temperature range (i.e. 0.5 to 5°C of global warming) but we expect yields to change similarly for similar warming. That's why the temperature-yield relation is not derived from one linear regression of the full scenario period (2005-2100) but from smaller subsets. These could be time slices of a fixed length which, however, would create a scenario dependence. Temperature bins combine yield changes from years with similar global warming, in our case years with global warming in a 0.5°C-wide range. For example, the 2°C bin contains all years with a global warming between 1.75 and 2.25°C. This allows to combine yield data independent of time (scenario), under the assumption that global mean temperature is the primary driver of long-term yield changes.*

Specific comments

Page 3, lines 100-105. I find it difficult to believe that spatially explicit crops models (driven with spatially explicit climate information even though it may require some kind of scaling) will be worse off than the approach presented in this manuscript.

*Reply: The main advantage of our approach compared to that in Blanc (2017) is that our approach does not require spatially explicit climate scenarios nor access to a crop model (and thus, considerably less computational resources). We will remove the statement in line 101 – 105 that using a two-step emulator approach (crop yield emulator as proposed in Blanc (2017) together with emulated climate) would lead to higher deviations than our one-step approach since we do not have concrete evidence for this.*

Page 4, lines 133-135. "The simulated impacts of climate and CO2 changes on global and regional crop yields are shown to be related to global mean temperature change, and to be largely independent of the emissions scenario". There is a bit of circular argument here and then later on in Figure 4 where most of the variance in crop yield is explained by crop models. I think, by choosing a specified ΔGMT you have made ΔGMT independent of

emission scenarios and ESMs. What remains scenario and ESM dependent is the year when this specified ΔGMT is reached. Results from climate models show that the typical spatial pattern of temperature change is also similar across ESMs – that is higher warming over land than over ocean and higher warming at high latitudes than in tropics. Combined with specified ΔGMT this means that it is somewhat expected that most of the variance in crop yield will be explained by crop models.

*Reply: The finding cited above may not be surprising, but is a necessary condition for the derivation of the emulators, which is the central part of our study. The goal of the emulators presented in this paper is to provide a method to quickly estimate yield changes for any level of global warming regardless of the specific emissions scenario (i.e. the timing of warming, but within the limits of maximum warming used to train the emulator). For this to work, yield changes need to be largely independent of the emissions scenario. Simulation results show that yield changes for a specific level of global warming depend mostly on the crop model and (to a lesser degree) the ESM which is why we derive crop model-specific and ESM-specific emulators.*

Section 2. page 4.) Please 1) specify the time period of projections, 2) introduce the four RCPs and what they imply (i.e. low, medium and high emissions scenarios), 3) introduce CMIP5 a bit more and 4) provide 1 or 2 sentence about how bias correction is done.

*Reply: Thank you, we will add more information on all the four mentioned points.*

Table 1. page 5. The description of "Fertilizer use" needs to be made more consistent across the five crop models. For example, for LPJ-GUESS I am not sure what does "no consideration of spatial and temporal changes in nutrient limitation" means. Does this mean nutrient constraints are not considered or a specified fertilizer application rate is assumed for all times and all parts of the world.

*Reply: The LPJ-GUESS model does not consider nutrient constraints on plant growth. Upon reflection, we have decided that much of the information in Table 1 is not directly relevant for our study: We develop and compare several emulator approaches to approximate yield changes simulated by the crop models. The differences between the crop models participating in ISIMIP have been studied elsewhere (e.g. Rosenzweig et al., 2014, Frieler et. al. 2017) and are not the focus of our study. Therefore, we will remove some of the information in Table 1 and refer the interested reader to the relevant literature. The remaining information in Table 1 will be clarified.*

Page 7, lines 200-221. What is MIRCA 2000? Is this an observation-based product?

*Reply: The MIRCA2000 dataset provides gridded growing area information for 26 individual crops or crop groups for the year 2000, distinguishing between rainfed and irrigated crops. It is based on a combination of remote-sensing-based and census-based information. The dataset is documented in Portmann et al., 2010 (cited on line 220).*

Figure 3 and for other similar figures. Please consider putting the figure titles (Maize, Rice, Soybeans and Wheat) in horizontal format and a bigger font size as suggested in the annotated manuscript. Also, as a reader I was wondering what are the limits on the horizontal colour bar. I found the colour bar a bit difficult to interpret. If the white colour represents yield change of less than ± 5% then how can light blue and light red colours represent changes of more than 50%. For example, +6% will likely be indicated by light blue colour but this is not more than 50%.

*Reply: Regarding font sizes we will check and improve readability of the figures. Regarding the colour legend there seems to be a misunderstanding: Red and blue colours denote the percentage of model combinations that agree on a change of at least 5%. White denotes cells where either less than 50% of model combinations agree on the sign of change or the change is less than 5%. A change of +6% will be indicated as either white, light blue or dark blue depending on the percentage of model combinations that agree on such change. This is explained in the figure caption. We will change the figure legend from a continuous colour bar to separate legend entries in order to avoid misunderstanding.*

Page 12, lines 333-334, "To quantify the extent of the CO2 induced scenario dependence and its potential reduction at each grid point . . .". I am unable to understand what do "CO2 induced scenario dependence" and "potential reduction" refer to. There are two possible aspects here and I am not sure which one is right in his context. The first is that of CO2 fertilization – is this what is being referred to in this sentence. Second, note that just like by choosing a specified ΔGMT of 2.5 degree Celsius the dependence on emissions scenario and ESM has been reduced ΔGMT is also related to change in atmospheric CO2 concentration. Is this what is being referred to here?

*Reply: We apologize for the confusion. We will rephrase this sentence to: "We use two methods to determine the CO2 effect on crop yields within each global mean temperature bin at each grid-cell".*

Page 13. Equations (1) and (2) are the crux of the paper. Yet, I unable to understand how these equations are used. Perhaps if more equations were used to describe each and every term of these equations it would have been easier to follow them. For example, the change in yield is actually a two-dimensional quantity (i.e. it depends on the geographical location) at a given time (represented by i in equations 1 and 2). Perhaps if (t) can be used to represent time and not as subscript and if these equations were written again properly it would be easier to understand the objective of these equations. I am also unable to appreciate if equation (1) and (2) were applied at each individual grid cell or to the time series of globally-summed yield. In this context, I am also unable to understand what do the different temperature bins refer to.

*Reply: Linear regressions according to Equation 1 and 2 are carried out for each grid cell and for each temperature bin separately. As mentioned in our first reply above, a temperature bin includes all years where the smoothed global mean temperature is within +-0.25°C of the respective bin temperature. E.g. the 2.5°C bin includes values between 2.25 and 2.75°C. Indices for grid cell and temperature bin are omitted from all variables in eq. (1) and (2) for the sake of visual clarity. We will improve the clarity of the text describing the equations and check if indices can be added to the equations themselves.*

In Figures 6 and 7, the absolute changes in yield are small (since they are mostly white in colour) yet the percentage changes are huge because those percentage changes are corresponding to small absolute values (as mentioned in the manuscript). Perhaps if the percentage changes can be masked over regions of low crop yields then these figures will be much more easier to interpret than the current percentage change figures. Alternatively, maybe the colour scale for figures with absolute yield changes can be changed and the percentage change figures can be removed.

*Reply: Thank you for the suggestion. We will use different colour scales for methods a and b (absolute changes) so that details will be better visible. We will move figures with relative changes to the supplementary online material.*

Page 18, lines 439-442. "While approach (b) requires a pair of crop model simulations – one with time-varying pCO2 and one with fixed pCO2, approach (a) only requires the default simulations with time-varying pCO2". I wasn't able to appreciate this. So perhaps this should be mentioned earlier on where the three emulator approaches are presented.

*Reply: We will outline the differences between emulator approaches earlier in the manuscript.*

Page 20, line 485-486. While shown in Lotze-Campen et al., 2008 please include the map of the 10 world regions in this manuscript as well.

*Reply: We will include a map of the 10 world regions.*

Page 20, line 486-487. "Compared to potential yields, using production gives less weight . . .". Please introduce (if necessary) and differentiate yield from production. Not all readers can be expected to appreciate this difference between the two.

*Reply: Yield refers to the harvest per unit area, e.g. tonnes per hectare. To derive production, yields are multiplied by the year-2000 harvested area. We will explain this better in the manuscript.*

Figure 10. It is very hard to see the thin lines corresponding to each RCP. Please consider including thick lines for say 8 or 10 year moving average values which would likely yield a more fair comparison with values from the emulators which do not account for the effect of climate variability on crop yield. Also, please consider using a different set of colours for the four RCPs. The green and the blue seem very similar and the orange and red are pretty close colours.

*Reply: In Figure 9, 10 and Table 1 (and the corresponding text in the second part of section 4), we compare decadal averages of regional crop production (crop yields multiplied by growing areas in the region) as simulated by the crop models with those calculated by the emulator methods. In Figure 10, these decadal values are marked by crosses (simulated) and circles (emulated production). Annual simulated production (thin lines) was added for illustrative purposes only. We will remove it in the revision, not least in order to avoid misunderstandings regarding the scope of the emulator (emulate long-term changes rather than inter-annual variability).*

Page 23, lines 541-544. I am unable to understand this. Also, I am unsure how there can be any variance attributed to CO2 effects. If this refers to the CO2 fertilization effect – doesn't CO2 changes gradually in all RCP scenarios. So yes, while there is a trend in specified CO2, there is not year-to-year variability. Or, am I misinterpreting this.

*Reply: We will rephrase this to "The variance is calculated separately for each RCP-GCM-GGCM combination and compared to the matching GCM-GGCM combination over the historical period (1980-2010)."*

I think, it should be made clear in the manuscript that the emulators do not capture the year-to-year variability but rather the long term trend in crop yield.

*Reply: We will add language to clarify this.*

---

## Author Comment (AC3) · 15 Feb 2018

**Response to**

**Anonymous Referee #2**

---The referee's comments are quoted in black, and our responses in *blue italics*.---

This paper succeeds in showing the relative contributions to the spread of crop yield projections from impact models, climate models and scenarios. Further, it also succeeds in demonstrating that there is potential utility to functions relating change in GMT to yield impacts in the future. Clear and precise outlining of methods and data availability makes the results of this paper easily traceable and its contents easily replicable by fellow scientists.
This paper would benefit from further discussion of methodological limitations. In particular, the following points should be addressed:

i) 5 GCMs are used to obtain climate projections. The authors should discuss the representativeness of these 5 GCMs with regards to the CMIP 5 ensemble.

*Reply: McSweeney and Jones, 2016 (http://dx.doi.org/10.1016/j.cliser.2016.02.001) investigated how representative the 5 ISIMIP GCMs are of the full CMIP5 ensemble. They find that the ISIMIP subset probably underestimates the total uncertainty in future climate impacts attributable to GCMs for many regions. On the other hand, they also conclude that it's not possible to represent much more of the total uncertainty with any other set of 5 GCMs; in other words, the ISIMIP subset is basically as good as one can get with only 5 GCMs. We will add discussion of this aspect to the revised paper.*

ii) Section 5 describes projected increases in regional crop yield variance. This section should also include a discussion of the extent of uncertainty present in these projections of rising variance.

*Reply: As far as the uncertainty can be quantified in terms of model agreement, our analysis serves this purpose by showing (in Fig. 11) the percentage of model combinations agreeing on a change in variance. In regions with light colours the uncertainty between models is higher than in dark-coloured regions, where most models agree on a change in variance. Regarding the ability of the models to realistically capture yield variability in the first place, we refer to more dedicated studies such as Frieler et al. 2017 (doi: 10.1002/2016EF000525) who quantify the portion of observed variability that can be reproduced with the GGCMs.*

The summary section should incorporate discussion of the above two points in relation to the strength of the conclusions drawn.

*Reply: Thank you for the suggestion. We will add discussion of these points to the revised manuscript.*

In addition to elaborating on methodological limitations, the following points require clarification and elaboration respectively:
Lines 393 – 394: The use of the word "likely" needs to be clarified here. Please provide a definitive answer as to whether or not very low present-day yield potential in these regions is leading to division by values close to zero.

*Reply: We will verify this again and rephrase if necessary.*

Lines 396 – 405: Please give an explanation, or hypothesis for the negative effects of $CO_2$ in the two "potentially important" regions mentioned.

*Reply: We have changed the regression for emulator method b) (equation 2) slightly to allow for a more robust determination of the CO2 effect and the additive correction term a0 in equation 2. The negative effects in the two "potentially important" regions were an artefact where the previous regression resulted in a high correction term and a negative effect. This is no longer the case in the fixed method.*

In terms of mathematical formulae, symbols, abbreviations and units, the following should be addressed:
Lines 169 onwards: Please use a clearer term for fixed CO2 than YnoCO2.

*Reply: We will consider using $Y_{fixedCO2}$ instead of YnoCO2 as a clearer term.*

Lines 340 and 350: Please describe what a1 represents in each equation in word form.

*Reply: a1 represents the magnitude CO2 fertilization effect in both equations. It is the part of the yield change attributable to the difference in atmospheric CO2, as opposed to yield change driven by changes in climate variables.*

Line 434: Please correct the use of <> in this equation.

*Reply: Thanks, we will correct this.*

In terms of changes to figures, the following should be addressed:

Figures 2,4,6,7 and 8: All of these figures need to be much larger to increase their readability. If possible, each figure should be on its own page.

*Reply: We will rescale these figures to the full page width in the revised layout (approx. 7 inches wide). This will make them larger than the current layout. We will also check all font sizes and increase them where necessary.*

Figure 5: This figure is only for the LPJML model, please explain the rationale for model selection or point to where other model results can be found.

*Reply: Whenever a figure shows results from just one GGCM we use LPJmL as an example in the main article. Alternative versions of these figures for the other GGCMs are available in the Supporting Information. However, it seems that we missed to provide alternative versions of Figure 5. We will add these, and apologize for the omission.*

---

## Author Response (AR1)

Dear Editor,

Please find enclosed our revised manuscript. We have addressed the reviewers' comments as outlined in the responses to the reviewers uploaded earlier. In some cases, restructuring of the manuscript and changes in results have meant that our text changes deviate somewhat from our earlier response (all changes are highlighted in the marked-up manuscript version).

Overall, we have restructured the paper and collected all methodical descriptions that had previously been scattered across sections 2, 3, and 4, now in one place in section 2. We have clarified the methods to better explain the differences between the three emulator approaches.

In the first submission, the emulators had been trained using only data from RCP8.5. After adding the significance test as suggested by the reviewers, and doing some more testing, we have decided to use all available data from all four RCPs to train the emulator. This has improved overall emulator performance and has led to extensive changes to the results in section 3.2 and 4. Results for the emulators trained only on RCP8.5 are still available in the Supplement, and the effect of using either the full or limited training data is discussed at several points in sections 3.2 and 4.

Moreover, we discovered that pDSSAT (one of the crop models) deviated from the simulation protocol which meant that we had to redo the analysis for this model. We have added descriptions of the differences between model settings of pDSSAT and the other models in section 2 and added text to the results sections where these differences in simulation setup lead to differences in the results.

Attached to this letter you will find a marked-up manuscript version showing all changes to the text. In addition to the text changes, we have also updated all figures. Figure updates include better colour scales, increased font sizes for the labels, and improved figure layouts to make better use of the available space.

We believe that the manuscript has improved substantially thanks to the reviews, and hope that it is now acceptable for publication in ESD. We look forward to your response, and remain at your disposal for any further questions.

For the authors,

Sebastian Ostberg, Jacob Schewe

[revised manuscript text omitted]
_{varCO2}(i,t) = \Delta Y_{clim}(i) + a_1(i) \cdot (pCO2(t) - 370ppm) + \epsilon(i,t), \tag{1}$$

where $\Delta Y_{varCO2}(i,t)$ is the absolute yield change in grid point $i$ and year $t$ with respect to the historical reference period (1980–2010) and $pCO2(t)$ is the atmospheric $CO_2$ concentration of the corresponding year. In this statistical model, $\Delta Y_{clim}(i)$ represents an estimate of the purely climate-induced yield change at the respective bin temperature, but assuming a fixed year-2000 $pCO_2$ of 370 ppm (i.e. without $CO_2$ fertilization), $a_1(i)$ represents the added effect of $CO_2$ fertilization, and $\epsilon(i,t) \backsim N(0,\sigma^2)$ represents the residual error.

**2.3.2 Approach (b)**

Approach (b) fits the following linear regression model to the yield difference between the default and fixed-$CO_2$ simulation for all years falling into a specific ΔGMT bin:

$$Y_{varCO2}(i,t) - Y_{fixedCO2}(i,t) = \\ a_1(i) \cdot (pCO2(t) - pCO2_{ref}) + \epsilon(i,t), \tag{2}$$

[revised manuscript text omitted]

**3.2 Direct impacts of increasing pCO2 $p\text{CO}_2$**

In addition to air temperature warming, pCO2 $p\text{CO}_2$ has a direct influence on crop yields. As it varies within the different $\Delta$GMT bins, it is expected to induce part of the fluctuations of the yield changes at given GMT levels. We find that this CO2 effect is not scenario dependent (see Figure 5 $\text{CO}_2$ effect shows little scenario dependence (see Figure 5 for the global average effect within the LPJmL simulations at $\Delta$GMT=2.5°C), consistent with a short response time of plants to pCO2 changes. $p\text{CO}_2$ changes. As expected, the $\text{CO}_2$-induced yield differences increase with heightened atmospheric CO2 $\text{CO}_2$ level under all emissions scenarios, implying a stronger CO2 $\text{CO}_2$ fertilization impact with increased pCO2. A least squares fit to the yield differences versus greenhouse gas level within each $\Delta$GMT bin allows for a quantification of the direct CO2 effect at each level of warming based on global pCO2, rather than the emissions pathway. The underlying assumption is that the effect of the temperature variation within the 0.5°C range of each $\Delta$GMT bin will be minimal compared to the effect of the CO2 variation across all RCPs pCO2.

[revised manuscript text omitted]

**4 Validation of three emulator approaches**

Based on the on yield from those of $pCO_2$ change. By design, climate-induced patterns (assuming fixed year 2000 levels of CO2) of relative yield changes and the associated within-bin relationship between CO2 and crop yields identified in section 3, we propose the following two-step interpolation method to compute crop yield changes for any given pair of $\Delta\text{GMT}$ and pCO2, using either of the above regression methods (a) or (b) : Linear interpolation between the temperature-specific, CO2-adjusted yield patterns of neighboring $\Delta\text{GMT}$ bins ($a_0(\Delta\text{GMT})$ from method 
[revised manuscript text omitted]

---

## Editor Decision (ED1)

[revised manuscript text omitted]

*[handwritten annotation: Are $\Delta Y_{clim}$ and $a_1$ the two variables/parameters determined by regression. Please clarify.]*

For all years falling into a specific $\Delta$GMT bin, approach (a) fits the following linear regression model to the response of yields in the default simulation to the increase in $pCO_2$:

$$\Delta Y_{varCO2}(i,t) = \Delta Y_{clim}(i) + a_1(i) \cdot (pCO2(t) - 370\text{ppm}) \tag{1}$$
$$+ \epsilon(i,t),$$

where $\Delta Y_{varCO2}(i,t)$ is the absolute yield change in grid point $i$ and year $t$ with respect to the historical reference period (1980–2010) and $pCO2(t)$ is the atmospheric $CO_2$ concentration of the corresponding year. In this statistical model, $\Delta Y_{clim}(i)$ represents an estimate of the purely climate-induced yield change at the respective bin temperature, but assuming a fixed year-2000 $pCO_2$ of 370 ppm (i.e. without $CO_2$ fertilization), $a_1(i)$ represents the added effect of $CO_2$ fertilization, and $\epsilon(i,t) \backsim N(0,\sigma^2)$ represents the residual error.

**2.3.2 Approach (b)**

Approach (b) fits the following linear regression model to the yield difference between the default and fixed-$CO_2$ simulation for all years falling into a specific $\Delta$GMT bin:

$$Y_{varCO2}(i,t) - Y_{fixedCO2}(i,t) =$$
$$a_1(i) \cdot (pCO2(t) - pCO2_{ref}) + \epsilon(i,t), \tag{2}$$

where $Y_{varCO2}(i,t)$ and $Y_{fixedCO2}(i,t)$ is the absolute yield in grid point $i$ and year $t$ of the default and fixed-$CO_2$ simu-

*[handwritten annotation: Pls clarify $a_1$ is determined by regression.]*

lation, respectively, $pCO2(t)$ is the atmospheric $CO_2$ concentration of the default simulation during the respective year and $pCO2_{ref}$ is the crop-model specific $pCO_2$ value of the fixed-$CO_2$ simulation (see Table 1). In this statistical model, $a_1(i)$ represents the $CO_2$ fertilization effect and $\epsilon(i,t) \curvearrowright N(0,\sigma^2)$ represents the residual error. No intercept is estimated in this model because yields from the default and fixed-$CO_2$ runs are expected to be identical if $pCO2(t) = pCO2_{ref}$. The purely climate-induced yield change at a fixed year-2000 $pCO_2$ of 370 ppm $\Delta Y_{clim}(i)$ can then be derived as:

$$\Delta Y_{clim}(i) = \Delta Y_{fixedCO2}(i) + a_1(i) \cdot (pCO2_{ref} - 370\text{ppm}), \quad (3)$$

where $\Delta Y_{fixedCO2}(i)$ is the average yield change in the respective warming bin of the fixed $CO_2$ simulation with respect to the historical reference period and $a_1(i) \cdot (pCO2_{ref} - 370\text{ppm})$ corrects for the different $pCO2_{ref}$ used by each GGCM.

**2.4 Emulator of temperature and $CO_2$ effects**

Based on the spatial patterns of purely climate-induced yield change $\Delta Y_{clim}(i)$ and added $CO_2$ fertilization effect $a_1(i)$, which are derived separately for each rain-fed and irrigated crop and specific to each crop model and GCM, we propose the following two-step interpolation method to compute crop yield changes for any given pair of $\Delta GMT$ and $pCO_2$, using either the coefficients from approach (a) or (b):

1. linear interpolation of $\Delta Y_{clim}(i)$ between the two neighbouring warming bins to the desired $\Delta GMT$ value, *temperature change*

2. addition of the $CO_2$ pattern described by $a_1(i) \cdot$ $(pCO2 - 370\text{ppm})$, where $a_1(i)$ is also interpolated linearly between the respective coefficients from the neighbouring warming bins, *temperature change*

The application of these two steps using coefficients from method (a) above will be called emulator approach (a); their application using coefficients from regression method (b) will be called emulator approach (b). In addition, we propose a third, very basic emulator approach (c) where the yield change for any given $\Delta GMT$ is derived from a simple linear interpolation of the average yield change in the neighbouring warming bins of the default simulations $\Delta Y_{varCO2}(i)$ with respect to the historical reference period, without using the associated $pCO_2$ as additional predictor.

The linear interpolation of any of the previous coefficients between two neighbouring warming bins is illustrated for a $\Delta GMT$ of 2.3°C as follows:

$$coef(i, 2.3°C) = (1 - \delta) \cdot coef(i, 2°C) + \delta \cdot coef(i, 2.5°C),$$

[revised manuscript text omitted]

*[handwritten at top: Need to say somewhere emulators do not have the ability to consider inter annual variability in crop yield.]*

the price, whose volatility is tightly linked to rapid changes in supply (Gilbert and Morgan, 2010).

**6  Summary**

Evaluating the impacts of climate change at different levels of global warming, and thus evaluating mitigation targets, requires a functional link between $\Delta GMT$ and regional impacts. Here we have shown that changes in crop yields, as simulated by gridded global crop models, can be reconstructed based on $\Delta GMT$, with some limitations. The small spread of simulated yield change across the RCP scenarios as compared to the GCMs and impact models implies that projected impacts at different $\Delta GMT$ levels are not substantially dependent on the choice of emissions pathway. In this context, it has to be noted that the scenario setup of the ISIMIP crop model simulations was chosen specifically to minimize scenario-dependency by asking modellers to keep crop management fixed at present-day level or adjust it only in response to climate without any regard to the time horizons associated with adaptation or economic processes. Four models are calibrated to match present-day yield levels while LPJ-GUESS simulates potential yields assuming optimal management. Only two of the crop models allow for an adjustment of planting dates in response to climate change (GEPIC and PEGASUS, see Table 1). Three of the models keep the total heat unit sum to reach maturity constant, assuming no change in crop cultivar which effectively leads to a shortening of the growing season. Representation of soil nutrient limitation varies substantially between models, with two models (LPJ-GUESS and LPJmL) considering no soil nutrient limitation at all, while the nutrients considered and the assumptions on fertilizer application differ between the other three models. The effects of these assumptions on yield changes simulated by the different crop models are not studied here since the focus of this study is on developing efficient emulators, but these assumptions inform both the simulated yield changes as well as the emulators which attempt to imitate the behaviour of the crop models. The results of the ISIMIP crop models have been studied in detail in Rosenzweig et al. (2014).

We have tested three different approaches for emulating crop yield change simulated by five GGCMs driven by HadGEM2-ES climate projections for four RCPs. All approaches rely on $\Delta GMT$ as the main predictor of yield change at the grid scale. Two of the approaches include $pCO_2$ as an additional predictor. An approach (a) attributing the yield variation within an individual $\Delta GMT$ bin of a simulation with varying $pCO_2$ solely to the change in $pCO_2$ shows the poorest overall performance. An approach (b) based on the difference between runs with and without direct $CO_2$ fertilization effects performs similarly well as a simple approach (c) using only $\Delta GMT$ as a single predictor. Considering the added complexity in approach (b) compared to (c),

the simple approach (c) appears in general preferable even though it may not provide the best result in all regions. While our tests indicate that the emulators perform better for some crop models than for others we strongly advise against relying solely on results from any one particular model, but instead to always consider the full range of uncertainty spanned by the GGCMs. Similarly, different GCMs still account for more than 15% of the total variance of the ISIMIP ensemble at $\Delta GMT=2.5°C$ in a number regions (Figure 4) which is why emulators should be constructed for all GCMs.

Given the availability of crop model simulations in the ISIMIP archive, emulators based on approach (a) and (c) could be constructed for all five GGCMs for the remaining four GCMs (IPSL-CM5A-LR, MIROC-ESM-CHEM, GFDL-ESM2M, NorESM1-M). Emulators based on approach (b) could only be constructed for LPJmL and pDSSAT (and PEGASUS if using only RCP8.5 for training). With its five GCMs, which were selected from the CMIP5 ensemble based primarily on data availability at the time, the ISIMIP subset likely underestimates the total uncertainty in future climate impacts attributable to GCMs for many regions, however, the ISIMIP subset essentially samples as much uncertainty as is possible with only 5 GCMs (McSweeney and Jones, 2016). *[handwritten margin note: so what's the message?]* The generally good performance of approach (c) suggests that simplified predictions of large-scale agricultural yields may not require additional crop model simulations with $CO_2$ levels held at a historical level if planning to extend the GCM coverage.

The impact model ensemble assembled in this study also indicates that the variability of crop yields is projected to increase in conjunction with increasing $\Delta GMT$ in many important regions for the four major staple crops. Such an increase in yield volatility could have significant policy implications by affecting food prices and supplies, although management assumptions as well as model-structural limitations of the GGCMs to account for crop stress factors may impact the models' ability to project future changes in variability.

The scalability of mean yields is conducive to the development of predictor functions relating $\Delta GMT$, or other aggregate climate variables readily available from simplified climate models (such as $pCO_2$) to regional or global mean crop yield impacts. This lays the groundwork for a further exploration of the economic impacts of climate change encountered at target warming levels or over policy relevant regions.

**Data availability.** The coefficients estimated with Equations 1 to 3 are available as a Supplement, along with supplementary figures and RMSE estimates, at https://doi.org/10.5281/zenodo.1194045. The GGCM simulations that the analysis in this paper is based on are available through https://esg.pik-potsdam.de/search/isimip-ft/, with additional documentation available on the ISIMIP website https://www.isimip.org/outputdata/caveats-fast-track/

---

## Author Response (AR2)

Dear Editor,

Please find enclosed our revised manuscript. We have added clarifications where you requested them, as well as making some minor text changes in a number of places.

For example, we explained a little more what we mean by "simple climate model" on page 2.

Regarding your question on section 2.2, we do indeed relate local yield change in every grid point to Delta_GMT which is a global quantity. The concept of relating local changes to a global indicator is introduced in section 1. We have added text to section 2.2 explaining that local yield change is of course dependent on local temperature change, but that local temperature change is itself related to Delta_GMT. The ANOVA described in section 2.2 calculates the yield variance in each grid point explained by the GGCMs, GCMs, and RCPs.

Further, we have added another note to the Summary section (page 17, line 5–6 of the marked-up manuscript) pointing out that the emulator does not consider interannual variability. We have also rephrased the sentence around the McSweeney and Jones (2016) reference.

Attached to this letter you will find a marked-up manuscript version showing all changes to the text.

We thank you for all your suggestions that have helped to improve the manuscript, and hope that it is now acceptable for publication in ESD. We look forward to your response, and remain at your disposal for any further questions.

For the authors,

Sebastian Ostberg, Jacob Schewe

[revised manuscript text omitted]